# CLIP Exhibits Improved Compositional Generalization Through Representation Disentanglement

## Abstract

Vision-language models (VLMs), such as CLIP, have shown promising Out-of-Distribution (OoD) generalization under various flavors of distribution shifts. Recent studies attempted to investigate the leading cause of this property. In this work, we target the same goal, but focus on a certain type of distribution shift, in which test images contain *unseen compositions* of attribute-object pairs, but with the objects and attributes being *individually* seen during training. The models are expected to classify those images into the composition classes, i.e. attribute-object pairs, and also into object classes by ignoring attributes. We carefully designed an authentic image test dataset consisting of attributes for objects that are unlikely encountered in the CLIP training data. We found that the compositions diversity in the training data, as measured by normalized mutual information between objects and attributes, has a significant effect on the improvement of compositional generalization in the CLIP models. We found that image/text representation disentanglement with respect to the composition constituents also plays a key role in the improved generalization of these models. We notice that larger training datasets could potentially trigger emergence of such a disentanglement, as the compositions are typically more diverse in such datasets. We validate this hypothesis through different representation disentanglement metrics, including Z-Diff, and explicitness scores for various CLIPs. Our findings reveal a correlation between better OoD performance and higher scores in these disentanglement metrics, suggesting that improved disentanglement potentially contributes to enhanced compositional OoD generalization in VLMs.

## 1 Introduction

In recent years, several studies (1; 2; 3; 4) have tried to explore the reasons behind the generalization capabilities of the Vision-Language Models (VLMs) such as the CLIP (5). In particular, Out-of-distribution (OoD) generalization, which is the ability of a model to generalize to unseen data distributions, have been thoroughly investigated. (5; 1) demonstrated that CLIP has much higher OoD generalization competency compared to the precedent vision models. In an attempt to study this emergent property of the CLIP, it was observed that the dataset diversity is the main driver factor that significantly impacts the model OoD performance (1). On the other hand, a contrasting viewpoint was put forth, suggesting that language supervision may play a crucial role in transferability of CLIP representation compared to the SimCLR representation (6). These discrepancies on the origin of OoD generalization capability of the vision-language models warrants more investigation and reconciliation.

It is noteworthy that the OoD generalization has several flavors, such as distribution shift (7), spurious correlation (8), and compositional generalization (9). These types of generalizations should be studied separately as they have distinct natures. In addition, a challenging aspect of benchmarking the OoD generalization of VLMs is that the training distribution is not completely known in certain datasets. For instance, the training dataset of the OpenAI CLIP has not been released, which makes designing a test set that has a truly different distribution from the training one challenging. We aim to address these issues in this paper, by focusing our attention on the compositional generalization in the single object setting, and designing an authentic test dataset to assess the training data characteristics and mechanisms in the models that lead to the OoD generalization.

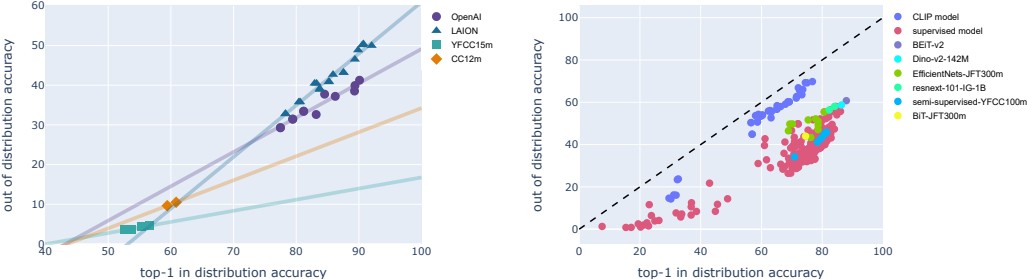

Figure 1: Left: Comparing effective compositional OoD generalization of CLIP models with diverse backbones and training sets in the zero shot setting, where no fine-tuning is performed on the target task. The in-distribution (ID) test set is the ImageNet validation split, with the labels being the object names, while the out-of-distribution test set is our designed compositional dataset, with labels being attribute-object pairs. Noticeably, there is a large gap between the performance of CLIPs that are trained on small datasets, e.g. CC12m and YFCC15m, and on gigantic datasets such as LAION and OpenAI. Right: Comparing OoD generalization of various models including CLIPs, supervised models, and the self-supervised models like DINO-v2 and BEiT-v2. ID and OoD test sets are the same as before, with the labels being the object names in both ID and OoD test sets, as the attributes are not among the labels of the pre-trained supervised models. Despite being competitive on ID accuracy, the supervised models fall short of the compositional OoD accuracy of the CLIP models.

Compositional OoD generalization is among the main branches of the OoD generalization, focusing specifically on the ability of models to generalize to unseen combinations of known concepts or entities. The VLMs ability in encoding the compositional relationships between objects and attributes has recently received attention (10; 11). Although (10; 12) discussed shortcommings of VLMs in encoding the compositional relationships between objects and attributes, (11) showed that VLMs can compose concepts in a single-object setting including single attribute-object compositions. Furthermore, some more recent work highlighted the role of caption quality and elaboration in enhancing the compositional reasoning in VLMs (13). In a nutshell, the literature suggests that compositional reasoning in VLMs might be more feasible in the single-object setups, and that is why we focus on this setting. Furthermore, most of the work around compositional reasoning in computer vision were more concerned about compositional understanding and formation of the inputs, and less attention has been paid to the OoD generalization in which the generalization ability of the models are evaluated against truly novel compositions.

Most of the mentioned literature have built their evaluations on top of visual compositional reasoning benchmarks such as Attribute-Relation-Order (ARO) (10) and VL-Checklist (14). The primary issue with these benchmarks is that they rely on the existing datasets such as the Visual Genome, and perturbing the captions, e.g. exchanging the order of tokens and setting the goal as to distinguish the correct vs incorrect captions. As a result, such benchmarks do not rule out training data contamination of the test set, which would become highly likely as the training set gets into the scale of billion images. Furthermore, most of the captions in such benchmarks are naturally occurring in real-world, often making their perturbation, or incorrect version, unrealistic. Hence, if the language encoder of a VLM is a good "world model," the model may distinguish such captions irrespective of the visual content in the input image.

To this end, we propose a new benchmark to evaluate the compositional OoD performance of the CLIP models. Our benchmark involves generating a new compositional OoD dataset that is distinct from the training data that has been used to train the CLIP. We consider unseen pairs of attribute and object as the basis of creating our images through text-to-image models and manually controlling the image quality, and authenticity to the text. By assessing the captions in the training sets, we guarantee that none of the captions in our test dataset or similar captions are included in the CLIP training data. We then evaluate different CLIP models on our OoD dataset to determine which models perform better on this type of data. Fig. 1 gives an overview of this result.

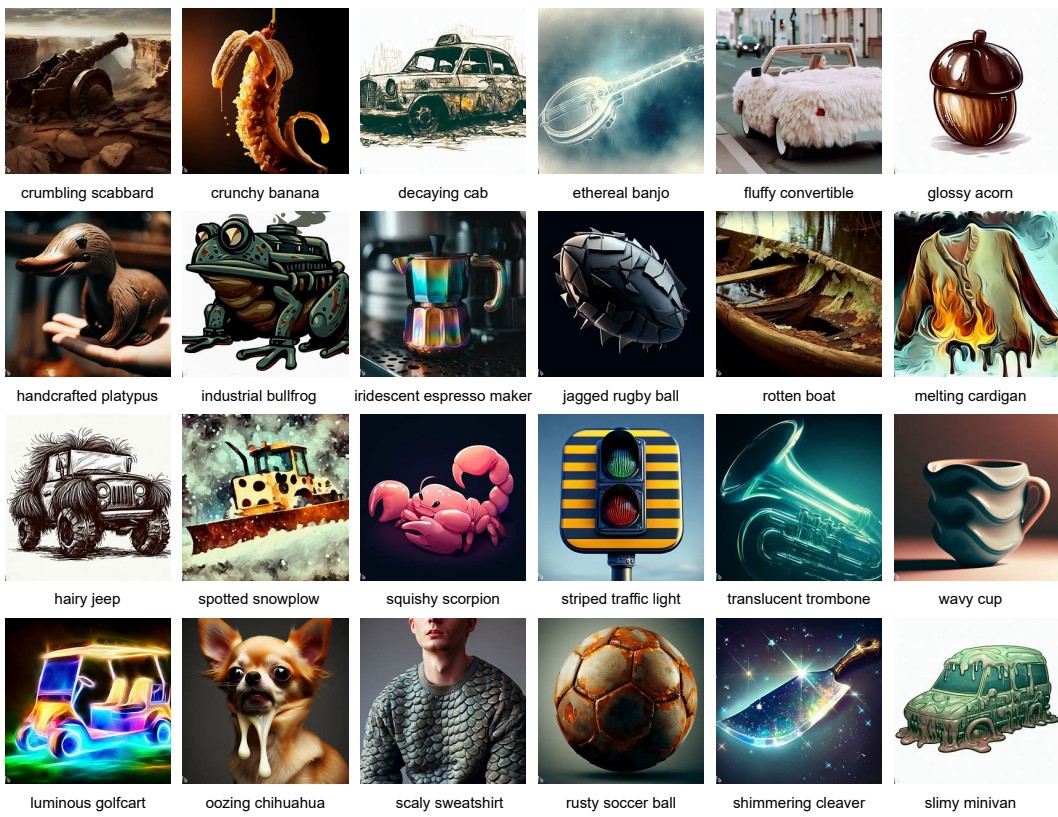

Figure 2: Examples of images from our generated dataset. This dataset is created by combining attributes and objects that do not appear in the CLIP training sets, specifically designed for benchmarking compositional OoD generalization purposes.

Finally, we analyze the factors that contribute to better performance in our benchmark, providing insights into how to improve the generalization capabilities of the CLIP models in the compositional setting. We observed that a lower normalized mutual information between the objects and their associated attributes in the training captions highly correlates with the compositional OoD generalization. We also found that the CLIPs that show higher OoD generalization typically exhibit strong disentangled text representations. Furthermore, such CLIPs also enjoy a more disentangled image representation with respect to the attributes and objects as well. Based on these results, we hypothesize that the compositionality of the language plays a key role in the improved generalization of the CLIP. Specifically, a dataset with diverse compositions of attribute-objects facilitates a more disentangled text representation, which in turn induces a disentangled image representation through contrastive learning. We elaborate on this hypothesis in Sec. 3.3.

Our contributions are summarized as follows:

- Designing an image test dataset of attribute-object pairs that are unseen in common CLIP training datasets.

- Benchmarking the compositional generalization of various CLIPs, and other models such as DINO-v2 (15), and BEiT-v2 (16), in the carefully designed and controlled setting.

- Identifying the necessity of compositional diversity of training captions for the CLIP to exhibit disentangled representation and therefore basic compositional generalization.

- Identifying representation disentanglement with respect to the variation factors, attribute and objects, as one of the key contributors to the CLIP improved compositional OoD generalization.

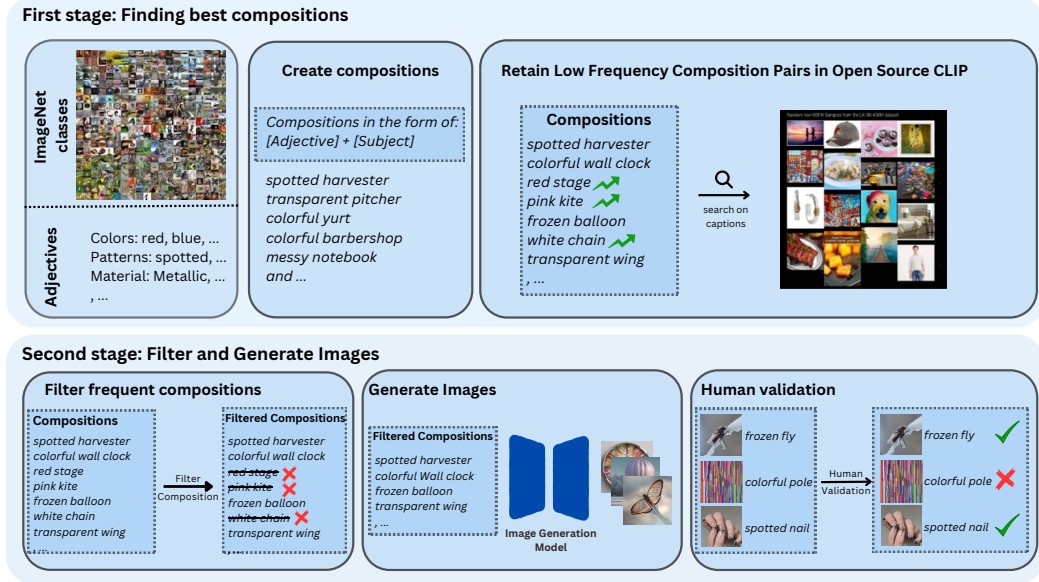

Figure 3: Diagram showing the process of generating our compositional benchmark, the ImageNet-AO dataset.

## 2 METHODOLOGY

In this section, we explain how we conducted our study step-by-step. We first describe how we created our challenging benchmark dataset, ImageNet-AO, which involves finding new combinations and making images with text-to-image models (Sec. 2.1). Examples of images in ImageNet-AO are shown in Fig. 2. Then, we dive into how we test CLIP models in the zero-shot setting, and the chosen criteria to evaluate the models (Sec. 2.2).

### 2.1 IMAGENET-AO: DATASET DESGIN

To effectively assess the compositional generalization capabilities of the models, we created a unique dataset of rare compositions, ensuring these were not present in the models' training data. This dataset was produced by creating compositional images via a text-to-image model, using an Attribute+Object template. An overview of the process is shown in Fig. 3, and outlined as follows:

**Selecting objects or nouns:** We extracted class names from the ImageNet dataset, using these as objects (or nouns) in our structure to create a link between the generated images and ImageNet classes. This allows for comparison of model performances on the familiar ImageNet validation set. We aimed for a diverse set of class names to enhance the complexity of the generated images.

**Selecting attributes or adjectives:** The next step involved choosing 30 distinct adjectives that were relevant and could create unique combinations with the selected nouns, enhancing the diversity of our compositional images.

**Selecting unseen (attribute, object) pairs:** We combined the 30 adjectives with a pool of 1000 nouns, resulting in 30000 distinct pairs. These were given to the text-to-image model to generate corresponding images. To ensure these combinations were not present in the CLIP training set, we conducted a thorough search and removed any combinations that were found.

**Generating images for (attribute, object) pairs:** The selected combinations were given to a text-to-image model for the image generation. Among various models, the Microsoft model powered by DALL-E proved to be the most powerful and was employed in our work.

**Validating the generated images:** Lastly, human evaluation was used to validate the generated images, with images not closely aligning with their prompts removed. After this process, around 12000 combinations remained, for which we successfully generated near 50000 accurate, high-quality im-

ages. An illustrative example of the diversified dataset generated through this process can be observed in Fig. 2. This figure showcases a subset of images that exhibit high degrees of alignment with their corresponding prompts, highlighting the effectiveness of the validation procedure.

Details regarding the Dataset design process can be found in section 6.2 of the Appendix.

## 2.2 MODEL/DATA ZOO AND EVALUATION CRITERIA

We evaluate various CLIP models with different training datasets and encoder architectures in our experiments (Sec. 3.1). The training datasets that are considered include OpenAI's private dataset, LAION, YFCC15m, CC12m, and DataComp. The backbone image encoders include ResNet50, ResNet101, ViT-B-32, ViT-B-16, and ViT-L-14. This allows us to analyze the impact of training data scale, and the choice of image encoder on the compositional generalization abilities of the CLIP. We assess the compositional generalization abilities of the CLIP models through several experiments including zero-shot evaluation, cross-modal retrieval, few-shot learning, and linear probe evaluation on ImageNet-AO. These experiments provide insights into how well CLIP encoders capture novel compositions under varying conditions such as the absence of fine-tuning data.

## 3 RESULTS AND ANALYSIS

In this section, our primary objective is to leverage the generated dataset to analyze our hypotheses, focusing on the impact of language supervision and representation decomposability on compositional OoD performance. Our aim is to investigate the relationship between different aspects of the training dataset, and the disentanglement of attributes and objects in text and image representation on one hand, and the CLIP OoD generalization on the other hand.

## 3.1 COMPARISON OF CLIP MODELS

To evaluate the CLIP model performance in the classification tasks, we adopted the evaluation method developed by (17), similar to the zero-shot evaluation approach described in (5). Our evaluation involves providing the model with the actual images and various captions, obtaining embeddings for both the images and texts, and calculating their cosine similarities. This allows us to estimate the relevance of the captions to the image content, similar to a classification task. Since we only had class labels (attribute-object pairs) for the images, and many captions could match the class label, 80 captions were created for each class using several templates, similar to the approach described in (5). We prepared embeddings for these captions and took their average to obtain a final embedding, which was used in the zero-shot evaluation. For the test sets, all 1000 classes of ImageNet were used as the in-distribution set and expanded the number of classes to approximately 12000 for the OoD set.

We evaluated the performance of various CLIP models, which are shown in Fig. 1. We observe that models with different architectures trained on LAION 400m and LAION 2B datasets achieved better performance than the model trained on the OpenAI dataset. We later show that the superiority of these models attribute to their image/text embedding disentanglement (see Table 2).

Furthermore, the results in Fig. 1 show that increasing the size of the training dataset positively impacts the performance of CLIP in both in- and out-of-distribution settings. As the dataset size grows, the CLIP models demonstrate improved accuracy across the board. This trend supports the idea that larger training datasets contribute to better performance in compositional out-of-distribution generalization for the CLIP. However, we observed an exception that is the CLIP model trained on YFCC15m performed worse than the CC12m model despite its larger training set. This result indicates that factors beyond the dataset size can influence the model behavior. Notably, prior research (6) has pointed out that the CC12m dataset possesses captions of higher quality that are more closely aligned with their corresponding images.

Additionally, our results deviated from those reported by (1). Specifically, the accuracy trends exhibited by the CLIP models trained on different datasets show notable distinctions. In that study, it was observed that when CLIP models were trained on small datasets like YFCC15m, the slope of the line fitted to their accuracy trends was consistent with the slope of the line fitted to the corresponding points of the CLIP models trained on larger datasets. However, in our experiments on ImageNet-AO,

we found that these points exhibited fitted lines with different slopes. This observation highlights the importance of further investigating compositional out-of-distribution generalization and the need to understand the impact of dataset variations on the CLIP model performance. Additional information regarding the performance of diverse CLIP models trained on various datasets can be found in Sec. 6.3.1 of Appendix.

## 3.2 COMPARISON WITH OTHER LEARNING APPROACHES

In this experiment, our main objective is to investigate the impact of language supervision on the performance of CLIP models compared to supervised, semi-supervised, and self-supervised models under compositional OoD settings. It is important to note that our intention is not to conduct a direct and fair comparison between the two approaches, but rather to explore whether the specific language supervision used during CLIP training could contribute to the improved OoD accuracy.

Since ImageNet-AO was based on ImageNet classes, we assumed the object names as the class labels for the evaluation purposes. This allowed us to assess the accuracy of supervised models trained on ImageNet, on ImageNet-AO. For evaluation of the CLIP models, we took a distinct approach. We generated captions for each data point using only the object name, deliberately removing any accompanying attributes. This modification aimed to align the evaluation of CLIP models more closely with the evaluation process of the supervised models.

In Fig. 1 (right), we present the evaluation results of 200 supervised models pretrained on ImageNet, comparing their in-distribution and OoD accuracy with various CLIP models. We also evaluate several state-of-the-art semi-supervised and self-supervised models including DINO-v2, BEiT-v2, and BiT which were pre-trained on large unlabeled datasets. The findings clearly demonstrate a remarkable trend: CLIP models trained on OpenAI's dataset, LAION, and DataComp consistently outperform the other models, specially in the OoD accuracy.

We interpret these findings as strong evidence that the inclusion of language supervision, particularly during CLIP training, positively impacts the model representation quality, hence making it possible to generalize to unseen compositions, despite the absence of such compositions in their training data. These results underscore the potential of CLIP models, specifically their ability to surpass supervised and unspuervised models in the compositional OoD generalization. More experiments including evaluation of few-shot and fine-tuned CLIPs are available in 6.3 in the appendix. The zero-shot evaluation performance of other vision-language models on Imagenet-AO is also presented in 6.3.7 .

## 3.3 WHY CLIP HAS COMPOSITIONAL GENERALIZATION?

Having established the superior performance of certain CLIPs in compositional generalization, we next try to investigate the reasons behind these observations.

More disentangled image representations that provide more decomposable embedding dimensions of objects and attributes facilitate compositional generalization. Such representations make meaningful construction of known concepts in the embedding space feasible. We hypothesize that large and diverse datasets reduce the dependency between attributes and objects, promoting a more disentangled representation of texts and images. Here, disentanglement means assignment of separate and independent embedding dimensions to different factors of variations, which in this case are the objects and attributes. Based on these insights, we posit that disentanglement is the key to the CLIP unseen compositional generalization. This claim is supported by two main arguments:

- Large and diverse datasets reduce entanglement between object and attribute tokens. In other words, they help to promote a more disentangled text representation at the data level (see Sec. 3.3.1).

- Text representation disentanglement is induced in the image encoding, due to implicit maximization of the mutual information of text and image representations through contrastive learning. We elaborate on this claim empirically in Sec. 3.3.2, and theoretically in Sec. 3.3.3.

Table 1: Normalized Mutual Information between the attributes and objects calculated for captions of some CLIP training sets. The domain of these random variables are defined based on the compositions present in our generated dataset.

| Dataset | Dataset Size | NMI |
|---------|--------------|--------|
| YFCC | 15m | 0.9390 |
| CC | 12m | 0.8903 |
| LAION | 400m | 0.8307 |
| LAION | 12m | 0.902 |
| LAION | 2B | 0.854 |

### 3.3.1 MUTUAL INFORMATION OF ATTRIBUTE-OBJECT TOKENS IN THE TRAINING CAPTIONS

We hypothesize that the utilization of datasets with disentangled structures during training plays a pivotal role in facilitating the learning of disentangled representations by models. To evaluate the degree of disentanglement in the training captions utilized by the CLIP, we conducted an analysis by measuring the normalized mutual information (NMI) between the object class and attribute tokens, whose domains are defined based on the captions in our generated dataset. The NMI is calculated based on the captions in datasets on which CLIP was trained, enabling us to gauge the level of disentanglement present in the training data.

The findings are depicted in Table 1, which demonstrates that the LAION 400m dataset exhibits a lower NMI value compared to the CC12m dataset. Similarly, the CC12m dataset displays a lower NMI value compared to the YFCC15m dataset. These observations are aligned with the outcomes of our previous experiments on compositional OoD generalization.

When the mutual information between two variables in a dataset is reduced, it indicates a diminished statistical dependence among those variables. In the context of disentanglement, this implies that the factors of variation within the dataset are less entangled or intermingled. Additionally, the low values of NMI emphasize the diversity of textual compositions in the dataset. This diversity is a crucial aspect for the CLIP to attain high performance in effectively handling the OoD scenarios.

### 3.3.2 DISENTANGLEMENT METRICS AND MODEL PERFORMANCE

Having observed the disentanglement at the *data level*, we next tried to assess whether this property has been induced from the data to the *representation level*. Hence, in the next experiment, we aim to assess the level of disentanglement in embeddings of various CLIP models. We utilize some common disentanglement metrics, namely the Z-Diff Score (18), DCI-Informativeness (19), Explicitness score (20), and DCIMIG (21) metrics, to quantitatively evaluate the models. These metrics are typically employed for supervised disentanglement assessment and require access to the latent factors of data. Since we have a compositional text specifying the attribute and the object for each image, we can consider two super latent factors corresponding to attributes and objects respectively.

We calculate these metrics for each CLIP model on the ImageNet-AO. Subsequently, we create a scatter plot (Fig. 4) to visualize the relationship between OoD accuracy and the disentanglement metrics. Each point in the plot represents a CLIP model, with the x-axis denoting the OoD accuracy and the y-axis representing the disentanglement metric. As observed in the plot, there is a discernible pattern where models with higher OoD accuracy tend to exhibit more disentangled representations. This empirical observation aligns with our initial hypothesis.

### 3.3.3 WHY DISENTANGLEMENT MAY ARISE IN CONTRASTIVE LEARNING?

We next try to give some theoretical insight on why and how the disentanglement emerges in the CLIP vision encoder. Several studies have shown the relation between minimizing the contrastive loss and maximizing the mutual information (22). Therefore, the CLIP training implicitly maximizes the mutual information between text and image encodings. We claim that disentanglement in the text representation, which was evidenced previously, may encourage disentanglement in the image encoding. To see this, let $y_1$ and $y_2$ be the text embeddings for the objects and attributes, respectively. Let $x_1$ and $x_2$ be the corresponding image embeddings. Assuming a decomposable text embedding

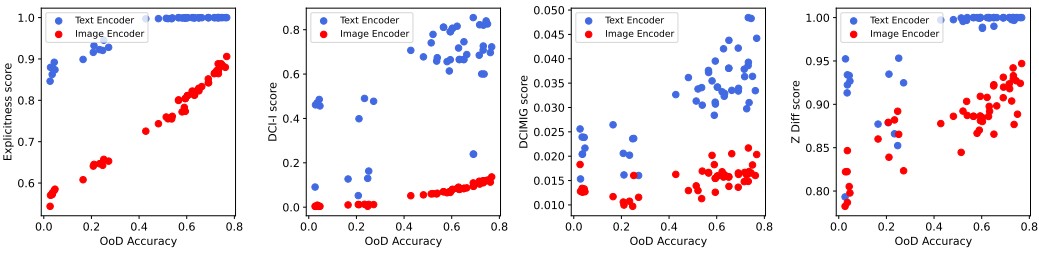

Figure 4: Disentanglment metrics vs. OoD Accuracy

means $y_1 \perp y_2$, i.e. $p(y_1, y_2) = p(y_1)p(y_2)$. Now by minimizing the contrastive loss, the mutual information $I(x_1, x_2; y_1, y_2)$ is maximized. By letting $x = (x_1, x_2)$, and $y = (y_1, y_2)$, we have:

$$
\begin{aligned}
I(x_1, x_2; y_1, y_2) &= D_{\text{KL}}(p(x, y) \parallel p(x)p(y)) \\
&= D_{\text{KL}}(p(x_1|x_2, y)p(x_2|y)p(y) \parallel p(x_1|x_2)p(x_2)p(y)) \\
&= \mathbb{E}_{x_1, x_2, y}[\log(p(x_1|x_2, y)/p(x_1|x_2))] + \mathbb{E}_{x_2, y}[\log(p(x_2|y)/p(x_2))] \\
&= \mathbb{E}_{x_2, y}[D_{\text{KL}}(p(x_1|x_2, y) \parallel p(x_1|x_2))] + \mathbb{E}_y[D_{\text{KL}}(p(x_2|y) \parallel p(x_2))]
\end{aligned}
$$

Maximization of the latter term makes $x_2$ and $y$ dependent random variables, otherwise if $x_2 \perp y$, the expected KL divergence would be minimum (or zero), which is against maximizing the mutual information. Note that however, $x_2$ does not ideally depend on both $y_1$ and $y_2$, otherwise the two distributions in the KL divergence in the first term become similar, which is also against maximizing the mutual information. Putting these together, $x_2$ mostly depends on $y_2$ if the mutual information is maximized. Using a symmetric argument, $x_1$ mostly depends on $y_1$. Finally, because $y_1 \perp y_2$, we conclude that $x_1$ and $x_2$ tend to become independent. Therefore, maximizing $I(x_1, x_2; y_1, y_2)$ decomposes $x$ if $y$ is already decomposed.

## 4 MORE ANALYSIS OF LEARNED EMBEDDINGS

### 4.1 DISENTANGLEMENT ANALYSIS OF CLIPs ON THE 3D SHAPES DATASET

Using the 3D Shapes dataset (23), we conducted two experiments to investigate the relationship between disentangled representations and OoD generalization in the CLIP models.

In first experiment, we employ the 480,000 images of 3D Shaped dataset, each with specific latent factors such as floor hue, wall hue, object hue, scale, shape and orientation. We train a classifier to calculate the Z-Diff Score and utilize it to determine which dimensions are most critical for each latent factor. In the process of calculating the Z-Diff score, we train a classifier that can determine, for a group of data points that have a fixed specific value for one of the latent factors,what that factor is. By using this classifier, we can identify which dimensions are more important for each factor.Subsequently, we extract the top 100 important dimensions for each factor and calculate how many dimensions are common across factors. Our results, presented in Table 2, demonstrate that models with higher OoD accuracy tend to exhibit fewer common dimensions across factors. This finding suggests that improved OoD generalization is associated with more disentangled representations.

In the second experiment, we looked at the impact of disentanglement on zero-shot object color manipulation using two identical images except for the object color. We calculated the embeddings using the CLIP and used the classifier of first experiment to identify the most important dimensions for detecting object color. By switching the top k dimensions, we tested the models' ability to detect captions matching the new color. The results are summarized in Table 2 showing that models with higher OoD accuracy require fewer dimension switches to achieve the color change, indicating that disentangled representations enable more effective zero-shot modifications.

Table 2: Number of common dimensions across factors and switching dimensions for color manipulation in 3D Shapes dataset

| Dataset | Architecture | OoD Accuracy | # of Common Dims | # of Switching Dims |
|---|---|---|---|---|
| LAION | ViT-L/14 | 43.14% | 2 | 40 |
| LAION | ViT-B/16 | 38.89% | 5 | 60 |
| LAION | ViT-B/32 | 32.84% | 7 | 90 |
| OpenAI | ViT-L/14 | 39.91% | 3 | 5 |
| OpenAI | ViT-B/16 | 37.73% | 4 | 10 |
| OpenAI | ViT-B/32 | 33.43% | 6 | 30 |
| CC | RN50 | 9.64% | 15 | 200 |
| YFCC | RN50 | 3.60% | 21 | 250 |

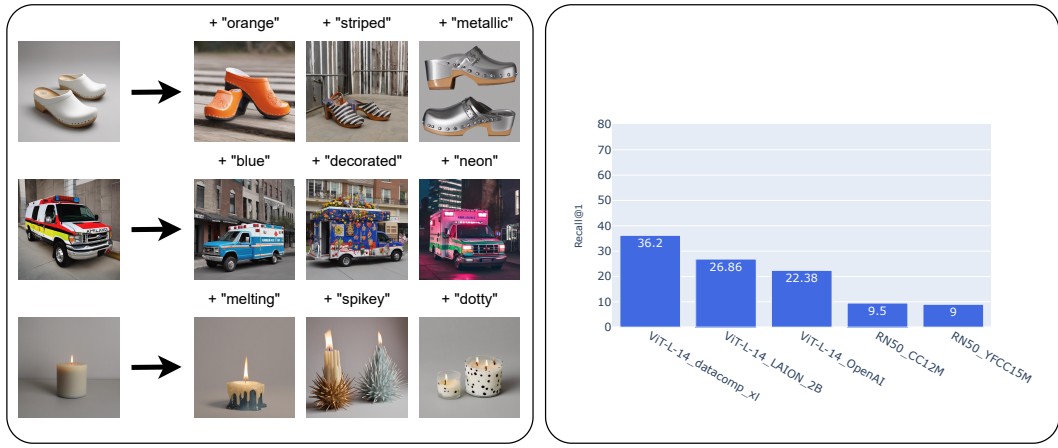

Figure 5: The performance of various CLIP models in the task of image±text retrieval. A model's superior performance in this task indicates that its representation is more disentangled.

## 4.2 IMAGE RETRIEVAL WITH IMAGE±TEXT QUERIES

Inspired by the work of (24), we designed an experiment to evaluate the compositional nature of embeddings learned by the CLIP models. Our primary objective is to assess the representation disentanglement of the CLIP models trained on diverse datasets. To accomplish this goal, we devised a test in which we input an image from our dataset into the image encoder of the model, and obtain its corresponding embedding. Next, we employed the text encoder of the model to compute the embedding of an adjective, ensuring that the adjective differed from those associated with the current image. These two embeddings were then combined through summation and used as a query in a process similar to the image retrieval. We then show the image closest to the generated query embedding. A total of 200 images were used to conduct this test for each model.

In order to evaluate the accuracy of the models predictions, we consider the image that is most similar to the query as the correct prediction if it possess both the intended object and adjective. A higher level of accuracy in the image retrieval task indicates that the model embeddings are more disentangled. Model evaluations are demonstrated in Fig. 5. The Recall@1 performance of various models aligns with our expectations. Specifically, we anticipated that models excelling in OoD tasks would also exhibit more disentangled representations. This highlights disentanglement as a possible mechanism towards better compositional generalization in the CLIPs.

## 5 CONCLUSION

This study examines how well CLIPs can generalize to new compositions of objects and attributes. We created an authentic benchmark of compositional images that are truly novel with respect to the CLIP training sets, and found that CLIPs ability to disentangle the text/images representation is

crucial for the compositional generalization. We showed that models trained on more diverse caption compositions perform better, and language supervision during training improves representation disentanglement. The study highlights the importance of dataset diversity and decomposability in enhancing vision-language models' compositional generalization capabilities.

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

# 6 APPENDIX

## 6.1 RELATED WORKS

### 6.1.1 ROBUSTNESS TO NATURAL DISTRIBUTION SHIFT

Maintaining the performance and generalization of machine learning models in the face of distribution shifts is a fundamental issue in real-world applications. In a study (25), researchers explored the relationship between the model's accuracy on the ImageNet, and its performance on a new dataset that was crawled and was attempted to be a replicate of ImageNet. This results in a minor and natural distribution shift in the test set. They found that while many models fail to maintain their original accuracy on the new test set, higher accuracy on ImageNet correlates with better performance on the new dataset. Therefore, the difference between the in-distribution and OoD accuracies reflects the *effective* OoD generalization that does not simply come from increasing the in-distribution accuracy.

Robustness under more intense real-world distribution shifts were investigated in image classification models' (26). This study tested 204 pretrained models on the ImageNet dataset under 213 different test conditions. It was discovered that the most effective way to improve the models' *effective robustness* is by training them on larger and more diverse training datasets. Vision-language models, such as CLIP, has shown improved effective OoD generalization in the zero-shot setting (5). Few shot fine tuning of such models were studied (27), which highlights the tradeoff between the effective robustness and the in-distribution accuracy. They examined various fine-tuning policies such as end-to-end or only the last layer to find the optimal balance.

The reason for the CLIP models' relative resistance to natural distribution shifts has also been investigated and explored in a similar work (1). In this work, by examining five possible factors, the training set size, the training distribution, language supervision at the training time, language supervision at the test time, and the contrastive loss function, their experiments revealed that the primary factor leading to improved robustness is a more diverse training distribution. However, in another study, it was empirically shown that the straightforward way of putting together various data sources to train the CLIP fails to enhance the effective OoD generalization, and in fact, diminishes the resilience of the most reliable individual data source (28).

### 6.1.2 COMPOSITIONAL GENERALIZATION OF CLIP

Compositional generalization refers to the capacity to generalize across unfamiliar compositions of familiar elements. Given that CLIP is a model that operates in a representation space encompassing both text and image, compositional generalization in the context of CLIP can be described as its capability to generalize to novel captions that contain familiar words. In this regard, recent studies have indicated that models like CLIP encounter difficulties when they need to engage in compositional reasoning (29). Specifically, they struggle in accurately associating the appropriate attributes with the corresponding objects, comprehending object relationships, systematically extrapolating to unfamiliar combinations of concepts, and handling larger and more intricate sentences (12; 30; 31).

Several innovative methodologies have been put forward to improve the efficiency of contrastive learning between images and text. Singh et al. (2023) (32), for instance, suggested an initial extraction of the scene graph from images, followed by the application of a graph decomposition and augmentation framework. This procedure aims to enhance coarse-to-fine contrastive learning. Bao et al. (2023) (33), on the other hand, employed Large Language Models (LLMs) to generate sentence-level descriptions for each compositional class. This approach involves breaking down visual language features into simpler primitives, resulting in a more precise compositional prediction. Nayak et al. (2023) (34) proposed a different strategy. They fine-tuned the vocabulary for attributes and objects on the seen classes to identify classes that compose tokens in a variety of ways, such as "old cat" and "white cat." When testing, they reassembled the learned attribute-object vocabulary in new combinations to recognize novel classes. Lastly, Yun et al. (2022) (35) focused their research on the spontaneous emergence of concept representations, like colors, shapes, or object attributes within the Contrastive Language–Image Pretraining (CLIP).

Our work differs from the previous work in that we focus on certain type of compositionality, i.e. object-attribute, in the single object setting. Here, our aim is to analyze existing CLIPs, rather than proposing new methods, to deal with compositionality. Furthermore, we provide various insights on

the root causes, possible mechanisms, and training data characteristics of CLIPs that improve this type of generalization.

## 6.2 DATASET DESIGN DETAILS

In order to generate compositional images, we devised a semi-systematic approach of providing input texts to a text-to-image model. Specifically, we employed a template consisting of 'Object + Attribute' to generate images depicting objects with attributes. In these prompts, we chose the attributes that are naturally unrelated to the corresponding objects. Instances of such prompts along with the generated images are shown in Fig. 2. This deliberate choice aimed to produce images with a significantly low probability of resembling any existing examples within the training dataset. Thus, to establish the pipeline for creating this dataset, we followed the following steps.

**Selecting objects or nouns.** In order to establish the initial steps for creating the dataset, we embarked on extracting the class names from the renowned ImageNet dataset. This vast collection of class labels serves as a comprehensive reference for a wide range of objects and concepts. By incorporating these class names as nouns in our predefined structure, we aimed to create a connection between the generated images and the classes present in the ImageNet dataset. This approach facilitated the subsequent evaluation and comparison of different models' performance on the familiar ImageNet validation set, which acts as a well-established in-distribution test set.

**Selecting attributes or adjectives.** In the second step of our pipeline, we focused on extracting 30 distinct attributes that were deemed highly relevant and useful in describing objects. Our aim was to identify attributes that result in novel and unique combinations with a significantly low probability of existence within the training dataset. By carefully selecting these attributes, we aimed to enhance the diversity and novelty of the generated compositional images with respect to the training data. These attributes are listed in the Appendix.

**Selecting unseen (object, attribute) pairs.** We proceeded to combine the selected 30 attributes with a pool of 1000 objects. This combination resulted in a total of 30,000 distinct pairs, which were now prepared to be fed into the text-to-image model for generating corresponding images. By leveraging this extensive range of object-attribute combinations, we aimed to produce a diverse and comprehensive dataset, showcasing a wide array of compositional variations for further analysis and evaluation.

Then, by matching these compositions to the image captions in the CLIP training set, we omit the object-attribute combinations that were present in the training dataset. This is essential to guarantee that the resulting dataset would truly be out-of-distribution, preventing the models from having encountered similar images during training. To achieve this, we conducted a thorough search through the training set captions of the CLIP models that will be evaluated. During this search, we employed a relaxed matching, meaning that if the object and attribute of a combination appeared within the captions but not in immediate proximity to each other, that particular combination would be removed from our list of valid combinations. This stringent process aimed to eliminate any potential overlap between the generated dataset and the training dataset, ensuring the novelty and uniqueness of the generated images. After completing the aforementioned search process, we removed approximately 20,000 combinations from our list. Here, we considered the captions in YFCC15m, CC12m, LAION 400m, and DataComp 128m as the training captions. We note that the remaining attribute-object pairs represent unusual *combination* of attributes and objects, while the attributes and objects are individually quite natural due to the initial selection of objects and attributes. Therefore, each object or attribute may have been encountered many times individually in the training set.

**Generating images for (object, attribute) pairs.** Moving forward, we aimed to utilize the selected combinations by providing them as input to the text-to-image model for the image generation. In our endeavor to generate novel images, we employed a powerful text-to-image models to explore various modeling approaches. Through this process, we evaluated different models, including Stable Diffusion (36), Stable Diffusion 2 (37), Deep Floyd, and the Microsoft model powered by DALL-E (38). Among these models, the Microsoft model proved to be the most powerful in generating images that are authentic with respect to the given composition. However, it should be noted that this model operates exclusively online and its model architecture and weights remain unpublished. Consequently, for each image we sought to create, we encountered the constraint of having to submit a request and patiently await the corresponding response. Despite its power, this model was not

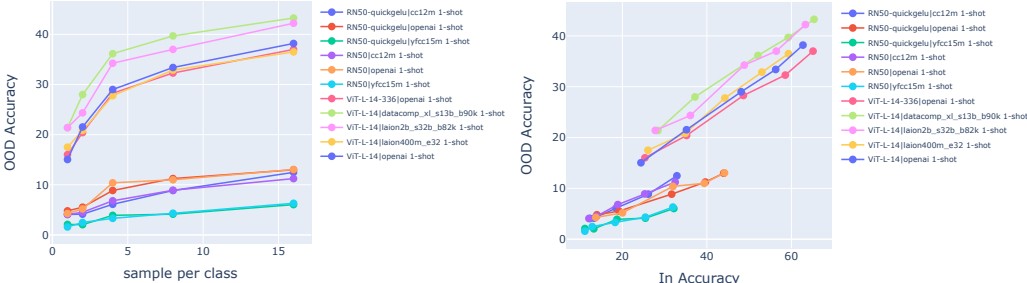

Figure 6: Comparison of OOD Accuracy in various few-shot settings for different CLIP models. This plot illustrates the out-of-distribution (OoD) accuracy performance across diverse few-shot scenarios, with the x-axis representing the number of samples used for fine-tuning, and the y-axis depicting the OoD accuracy.

without its issues. Notably, when the combinations contained animal names or in certain prompts, the model faced difficulties in generating valid images. Additionally, some prompts were blocked, and we were unable to discern the underlying reason for this occurrence.

**Validating the generated images.** Following the data crawling process, human supervision became crucial for validating the images generated by the model. This validation involved a meticulous examination of each image, assessing its proximity to the corresponding prompt. Any images that did not closely align with their respective prompts were promptly removed from the dataset. Consequently, after this rigorous manual validation process, only approximately 12,000 combinations remained, for which we successfully generated high-quality images that accurately corresponded to their given prompts. This careful curation ensured that the final dataset comprised compositions whose corresponding images carry their intended meanings. Fig. 3 illustrates the entire process of designing this dataset.

In the process of designing ImageNet-AO and ImageNet-AO-v2, we employed the following attributes: shimmering, oozing, fluffy, twisted, jagged, melting, furry, iridescent, translucent, bizarre, mangled, grotesque, wriggling, ethereal, crunchy, slimy, stretchy, sticky, rusty, decaying, hairy, wavy, crumbling, pulsating, rotten, scaly, luminous, squishy, spotted, striped, colorful, twisting, spiky, shiny, velvety, matte, polished, dotty, transparent, faint, neon, woody, chaotic, packed, decorated, messy, metallic, plastic, frozen, and a vibrant palette of green, red, yellow, blue, orange, purple, black, white, brown, and pink.

### 6.3 ADDITIONAL EXPERIMENTS

#### 6.3.1 IMAGENET-AO EVALUATION

The detailed zero-shot evalutions of various CLIP models on ImageNet-AO are presented in Table 3.

#### 6.3.2 FEW-SHOT EVALUATION OF CLIPs

In this assessment, we perform a few-shot evaluation of different CLIP models on Imagenet-AO. The main goal is to fine-tune a head for these models using a small number of samples per class: 1, 2, 4, 8, and 16, and then evaluate their performance. Few-shot learning is a crucial capability of CLIP, enabling the model to effectively generalize from limited training examples. During our few-shot evaluation, as depicted in Fig. 6, we noticed a similar trend to the zero-shot setting, in which again the CLIPs trained with DataComp and LAION datasets outperform the OpenAI CLIP.

Table 3: Evalution of various CLIP models on ImageNet-AO

| Model | Pretraining Dataset | ImageNet | ImageNet-AO |
|---|---|---|---|
| RN50 | OpenAI | 0.56928 | 0.292871 |
| RN50 | YFCC15M | 0.29564 | 0.036080 |
| RN50 | CC12M | 0.32314 | 0.096481 |
| RN50-quickgelu | OpenAI | 0.56928 | 0.292871 |
| RN50-quickgelu | YFCC15M | 0.30184 | 0.035950 |
| RN50-quickgelu | CC12M | 0.32626 | 0.105094 |
| RN101 | OpenAI | 0.58676 | 0.314501 |
| RN101 | YFCC15M | 0.31130 | 0.043326 |
| RN101-quickgelu | OpenAI | 0.58676 | 0.314501 |
| RN101-quickgelu | YFCC15M | 0.31904 | 0.046255 |
| RN50x4 | OpenAI | 0.63164 | 0.326130 |
| RN50x16 | OpenAI | 0.67862 | 0.371887 |
| RN50x64 | OpenAI | 0.71456 | 0.385208 |
| ViT-B-32 | OpenAI | 0.60306 | 0.334353 |
| ViT-B-32 | LAION-400M | 0.56548 | 0.328495 |
| ViT-B-32 | LAION-400M | 0.56598 | 0.327454 |
| ViT-B-32 | LAION-2B | 0.62270 | 0.404539 |
| ViT-B-32 | LAION-2B | 0.62866 | 0.404647 |
| ViT-B-32-quickgelu | OpenAI | 0.60306 | 0.334353 |
| ViT-B-32-quickgelu | LAION-400M | 0.59370 | 0.357741 |
| ViT-B-32-quickgelu | LAION-400M | 0.59454 | 0.357633 |
| ViT-B-16 | OpenAI | 0.65130 | 0.377311 |
| ViT-B-16 | LAION-400M | 0.63150 | 0.388961 |
| ViT-B-16 | LAION-400M | 0.63314 | 0.391478 |
| ViT-B-16 | LAION-2B | 0.66318 | 0.426777 |
| ViT-B-16-plus-240 | LAION-400M | 0.65036 | 0.409095 |
| ViT-B-16-plus-240 | LAION-400M | 0.65098 | 0.408119 |
| ViT-L-14 | OpenAI | 0.72592 | 0.399115 |
| ViT-L-14 | LAION-400M | 0.69078 | 0.432136 |
| ViT-L-14 | LAION-400M | 0.69170 | 0.431441 |
| ViT-L-14 | LAION-2B | 0.71558 | 0.465026 |
| ViT-L-14-336 | OpenAI | 0.73600 | 0.412523 |
| ViT-H-14 | LAION-2B | 0.74454 | 0.501410 |
| ViT-g-14 | LAION-2B | 0.73300 | 0.488501 |
| ViT-g-14 | LAION-2B | 0.75030 | 0.503840 |
| ViT-bigG-14 | LAION-2B | 0.76782 | 0.499609 |

Table 4: Models performance on text-to-image retrieval task

| Model | Pretraining dataset | R@1 | R@5 | R@10 |
|---|---|---|---|---|
| RN50 | OpenAI | 0.1628 | 0.4022 | 0.5318 |
| RN50 | YFCC15M | 0.0359 | 0.0995 | 0.1484 |
| RN50 | CC12M | 0.0627 | 0.1823 | 0.2673 |
| RN50-quickgelu | OpenAI | 0.1628 | 0.4022 | 0.5318 |
| RN50-quickgelu | YFCC15M | 0.0394 | 0.1076 | 0.1569 |
| RN50-quickgelu | CC12M | 0.0687 | 0.1918 | 0.2774 |
| RN101 | OpenAI | 0.1856 | 0.4349 | 0.5708 |
| RN101 | YFCC15M | 0.0404 | 0.1170 | 0.1670 |
| RN101-quickgelu | OpenAI | 0.1856 | 0.4349 | 0.5708 |
| RN101-quickgelu | YFCC15M | 0.0431 | 0.1233 | 0.1767 |
| ViT-B-32 | OpenAI | 0.2011 | 0.4674 | 0.6020 |
| ViT-B-32 | LAION-400M | 0.2161 | 0.4818 | 0.6109 |
| ViT-B-32 | LAION-400M | 0.2158 | 0.4803 | 0.6097 |
| ViT-B-32 | LAION-2B | 0.2748 | 0.5700 | 0.7058 |
| ViT-B-32 | LAION-2B | 0.2849 | 0.5751 | 0.7107 |
| ViT-B-32-quickgelu | OpenAI | 0.2011 | 0.4674 | 0.6020 |
| ViT-B-32-quickgelu | LAION-400M | 0.2437 | 0.5136 | 0.6468 |
| ViT-B-32-quickgelu | LAION-400M | 0.2416 | 0.5158 | 0.6474 |
| ViT-B-16 | OpenAI | 0.2313 | 0.5105 | 0.6533 |
| ViT-B-16 | LAION-400M | 0.2727 | 0.5654 | 0.6943 |
| ViT-B-16 | LAION-400M | 0.2754 | 0.5637 | 0.6921 |
| ViT-B-16 | LAION-2B | 0.3006 | 0.5964 | 0.7242 |
| ViT-L-14 | OpenAI | 0.2818 | 0.5864 | 0.7243 |
| ViT-L-14 | LAION-400M | 0.3305 | 0.6298 | 0.7548 |
| ViT-L-14 | LAION-400M | 0.3310 | 0.6304 | 0.7543 |
| ViT-L-14 | LAION-2B | 0.3790 | 0.6980 | 0.8108 |
| ViT-H-14 | LAION-2B | 0.3659 | 0.6652 | 0.7785 |
| ViT-g-14 | LAION-2B | 0.3628 | 0.6562 | 0.7720 |
| ViT-g-14 | LAION-2B | 0.3653 | 0.6542 | 0.7726 |
| ViT-bigG-14 | LAION-2B | 0.3757 | 0.6711 | 0.7893 |

### 6.3.3 EVALUATION OF FINE-TUNED CLIPs

In addition to zero-shot and few-shot assessment of CLIP models, we evaluated the image encoders of CLIP models that were fine-tuned on the full ImageNet dataset. The available image encoders fine-tuned this way were from CLIP OpenAI and CLIP LAION models. As shown in Fig. 5, we observed a similar trend to previous experiments - the model trained on a larger dataset with a larger backbone demonstrated better performance in the evaluation. Specifically, the CLIP LAION image encoder, which was trained on a larger LAION-400M dataset using a larger ViT-L/14 backbone, outperformed the CLIP OpenAI image encoder, which utilized a smaller ViT-B/16 backbone trained on OpenAI's smaller proprietary dataset. This finding further supports that models trained on larger datasets and with increased capacity tend to achieve improved compositional generalization.

### 6.3.4 TEXT-TO-IMAGE RETRIEVAL EVALUATION OF CLIPs

In this section, we delve into the text-to-image retrieval task and present a thorough evaluation of various CLIP models on ImageNet-AO. The objective of this evaluation is to examine how effectively each CLIP variant can retrieve relevant images based on textual queries, showcasing their ability to bridge the modal gap between language and vision. These results are shown in Table 4.

### 6.3.5 RELATION OF OUR COMPOSITIONAL SHIFT TO OTHER DOMAIN SHIFTS

In this section, we delve into the evaluation on different distribution shifts, where we rigorously assess the performance of various CLIP models across distinct types of ImageNet datasets. Specifically, we evaluate the models on ImageNet-A, ImageNet-R, ImageNet-Sketch, as well as Imagenet-

Table 5: Performance on a set of CLIP models on datasets showing various domain shift on ImageNet

| Model | Pretraining Dataset | ImageNet | ImageNet-v2 | Imagenet-sketch | ImageNet-R | ImageNet-A | Imagenet-AO |
|-------|---------------------|----------|-------------|-----------------|------------|------------|-------------|
| ViT-H-14 | LAION | 88.6 | 80.11 | 65.31 | 66.44 | 75.013 | 61.45 |
| ViT-L-14 | OpenAI | 88.3 | 80.33 | 63.79 | 65.64 | 77.64 | 61.87 |
| ViT-H-14 | LAION | 88.2 | 79.24 | 65.77 | 66.56 | 69.91 | 62.28 |
| ViT-L-14 | LAION | 88.2 | 78.87 | 59.74 | 59.74 | 68.84 | 59.6 |
| ViT-L-14 | OpenAI | 88.2 | 79.07 | 61.83 | 61.4 | 71.12 | 61.37 |
| ViT-L-14 | OpenAI | 87.9 | 79.26 | 62.52 | 63.47 | 70.85 | 61.39 |
| ViT-L-14 | LAION | 87.9 | 78.35 | 63.3 | 63.91 | 61.7 | 59.94 |
| ViT-H-14 | LAION | 87.6 | 79.06 | 67.94 | 68.05 | 64.76 | 62.72 |
| ViT-L-14 | LAION | 87.3 | 77.16 | 63.49 | 63.08 | 52.36 | 60.24 |
| ViT-B-16 | LAION | 87.2 | 77.77 | 53.09 | 49.45 | 58.48 | 56.41 |
| ViT-B-16 | OpenAI | 87 | 77.32 | 50.54 | 48.28 | 57.76 | 55.15 |
| ViT-B-16 | LAION | 86.6 | 77.51 | 56.42 | 53.03 | 54.09 | 57.27 |
| ViT-B-16 | OpenAI | 86.2 | 76.44 | 52.52 | 49.8 | 54.26 | 54.91 |
| ViT-B-16 | LAION | 86.2 | 75.53 | 52.09 | 49.17 | 46.88 | 55.9 |
| ViT-B-16 | OpenAI | 85.9 | 74.79 | 49.51 | 46.91 | 46.66 | 54.46 |
| ViT-B-32 | LAION | 85.8 | 75.55 | 47.74 | 44.78 | 50.92 | 53.57 |
| ViT-B-16 | LAION | 85.5 | 74.92 | 55.53 | 52.03 | 40.74 | 56.22 |
| ViT-B-32 | LAION | 85.4 | 75.08 | 48.36 | 45.29 | 46.61 | 53.35 |
| ViT-B-16 | OpenAI | 85.3 | 74.43 | 51.53 | 48.47 | 43.54 | 54.1 |
| ViT-B-32 | OpenAI | 85.2 | 74.22 | 45.96 | 42.92 | 42.413 | 52.23 |
| ViT-B-32 | LAION | 83.3 | 70.36 | 46.8 | 42.12 | 28.58 | 52.55 |
| ViT-B-32 | LAION | 82.6 | 69.26 | 49.52 | 43.9 | 21.81 | 51.51 |
| ViT-B-32 | OpenAI | 81.9 | 68.5 | 44.82 | 40.04 | 20.6 | 50.09 |

AO. Each of these datasets introduces specific domain shifts and challenges that differ from the standard ImageNet distribution. The results have been shown in Table 5. Moreover, for each pair of the datasets (i.e., domain shifts), the Kendall rank correlation between results of different methods on the corresponding datasets are presented in Fig. 7 demonstrates that ImageNet-AO shares higher similarity to ImageNet-R and ImageNet-Sketch when compared to the other datasets.

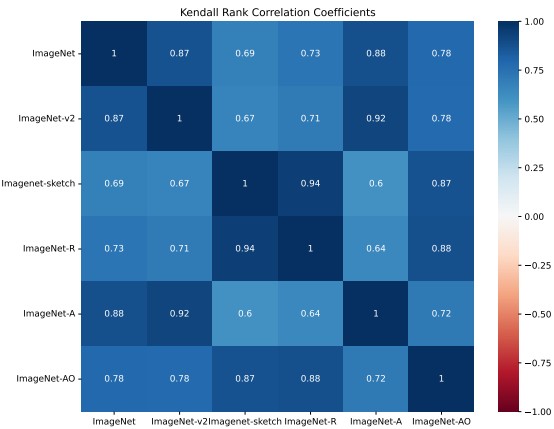

Figure 7: Kendall rank correlation on different dataset.

### 6.3.6 ZERO SHOT EVALUATION ON VARIANTS OF THE IMAGENET DATASET

We evaluate various CLIP models on different versions of the ImageNet dataset, including ImageNetV2, ImageNet-Sketch, ImageNet-R, and ImageNet-A. Our goal is to analyze the performance trends of models on these variant datasets and examine whether they correlate with results on our generated Imagenet-AO dataset. These results have shown in Fig. 9

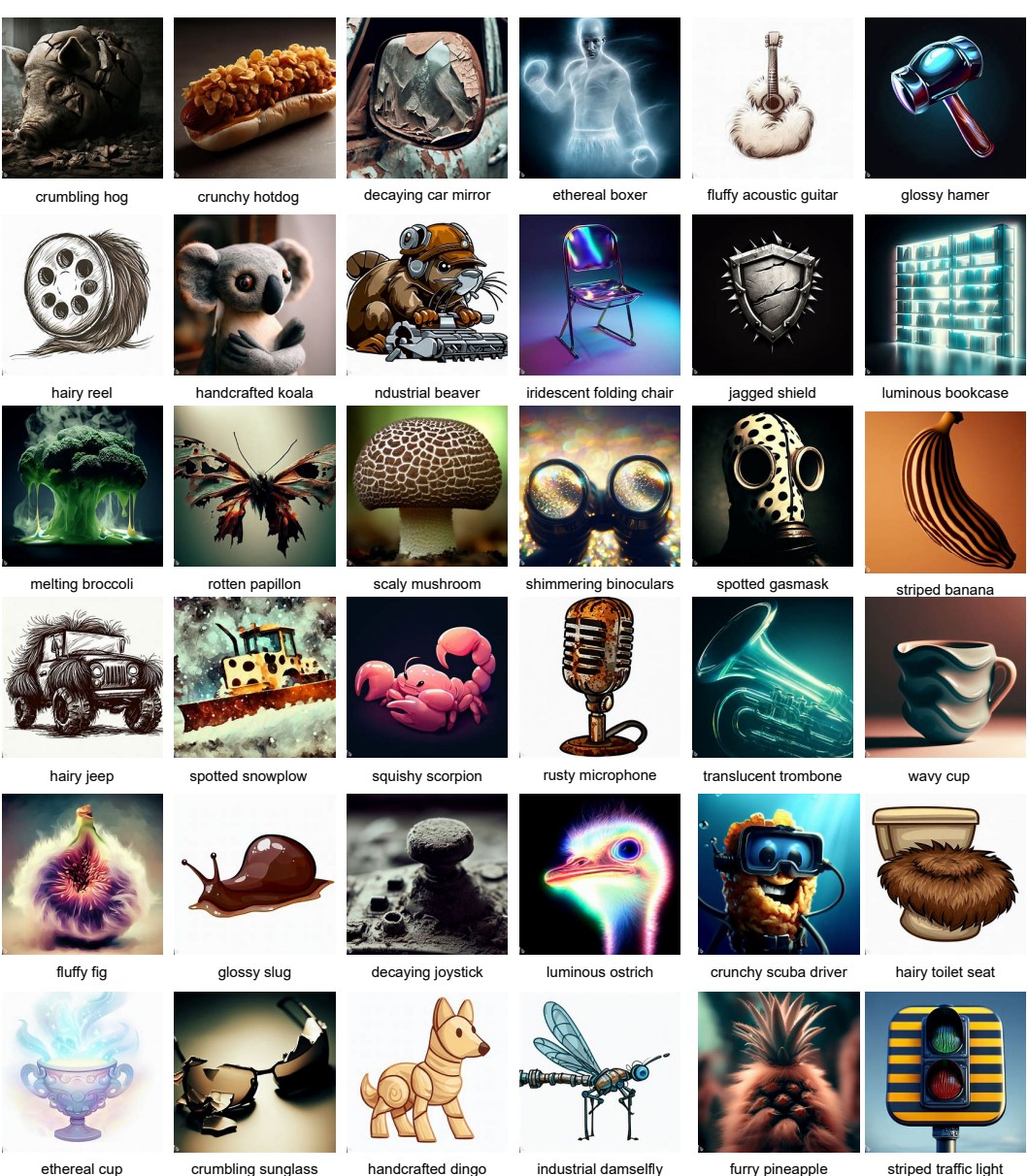

Figure 8: More examples of images from Imagenet-AO dataset.

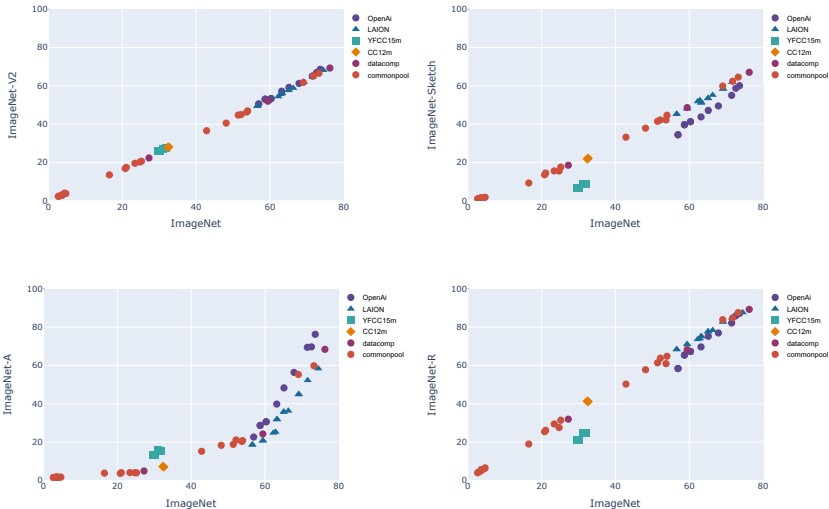

Figure 9: Performance of various CLIP models on versions of ImageNet with different domain shits vs. in-distribution ImageNet.

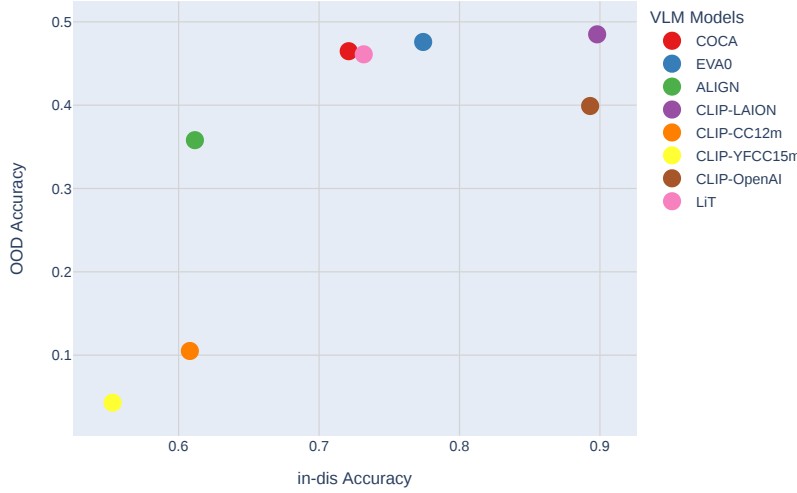

Figure 10: Zero-shot evaluation of different VLMs on ImageNet-AO

### 6.3.7 ZERO SHOT EVALUATION OF OTHER VLMS

We also evaluated other vision language models on ImageNet-AO using the settings from previous experiments. The results can be seen in Figure 10. As shown, other VLMs such as EVA (39) and Coca (40) demonstrated similar performance on ImageNet-AO compared to CLIP models.

### 6.4 OBJECT-OBJECT COMPOSITIONS

In the previous sections, we evaluated the CLIP models on specific composition types involving object-attributes. However, models may have abilities in handling object compositions too. To as-

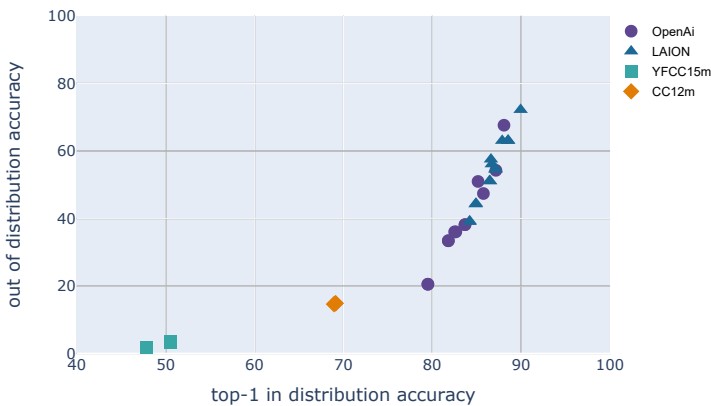

Figure 11: Object-object zero-shot evalution on the DomainNet dataset

sess this, we focus our attention on object-object compositions using the DomainNet dataset. We generate a new dataset of object-object image compositions by concatenating two random object images from DomainNet along the width. This creates compositions with two labeled objects. We evaluate CLIP models on this object-object dataset to analyze how well they can recognize both objects within a single image. The prompt template is "A photo of a [first object] and a [second object]". Accuracy is measured as correctly predicting both object labels. This analysis provides additional insight into how well the CLIP models comprehend relationships between multiple objects compared to individual objects. By using DomainNet as a source for the object images, we ensure diversity in the object types. The performance trends of the different models on object-object compositional generalization closely match their performance trends on attribute-object compositional generalization, as illustrated in Fig. 11. The x-axis of the scatter plot shows the model accuracy on DomainNet images, while the y-axis shows the model accuracy on our object-object images.

## 6.5 IMAGE-AO-V2

Creating images with novel compositions may cause a bias that images will be in the style of sketch and clipart. To overcome this issue, it is necessary to consider the rare compositions instead of just novel compositions and generate image according to these compositions. Therefore, we repeated the process of building the dataset as before, except that we kept the less frequent combinations. We used the SDXL model (41) to generate images for these rare compositions of attribute and objects. As opposed to novel compositions for which only Bing model is successful in generating images, for the rare compositions, open source models like SDXL are also helpful and more easily used to generate the dataset. In the human evalution section, we also removed invalid image, painting, and clipart styles. Figure 13 shows some examples of the final dataset. The evaluation results of different models can also be seen in Figure 12. As can be seen from the figure, the general trend of the results is the same as the previous experiments and the only difference is in the slope of the fitted line in some models.

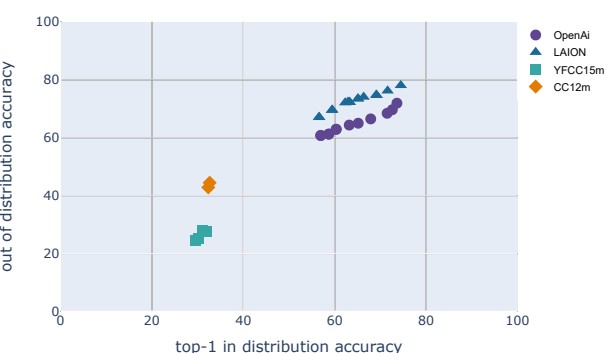

Figure 12: Comparing zero-shot OoD generalization on low-frequent compositional concepts for CLIP models with diverse backbones and training sets. Similar to our experiments in Sec. 3.1, ImageNet validation split with object labels is considered as in-distribution dataset. Out-of-distribution is an attribute-object dataset with infrequent compositional labels.

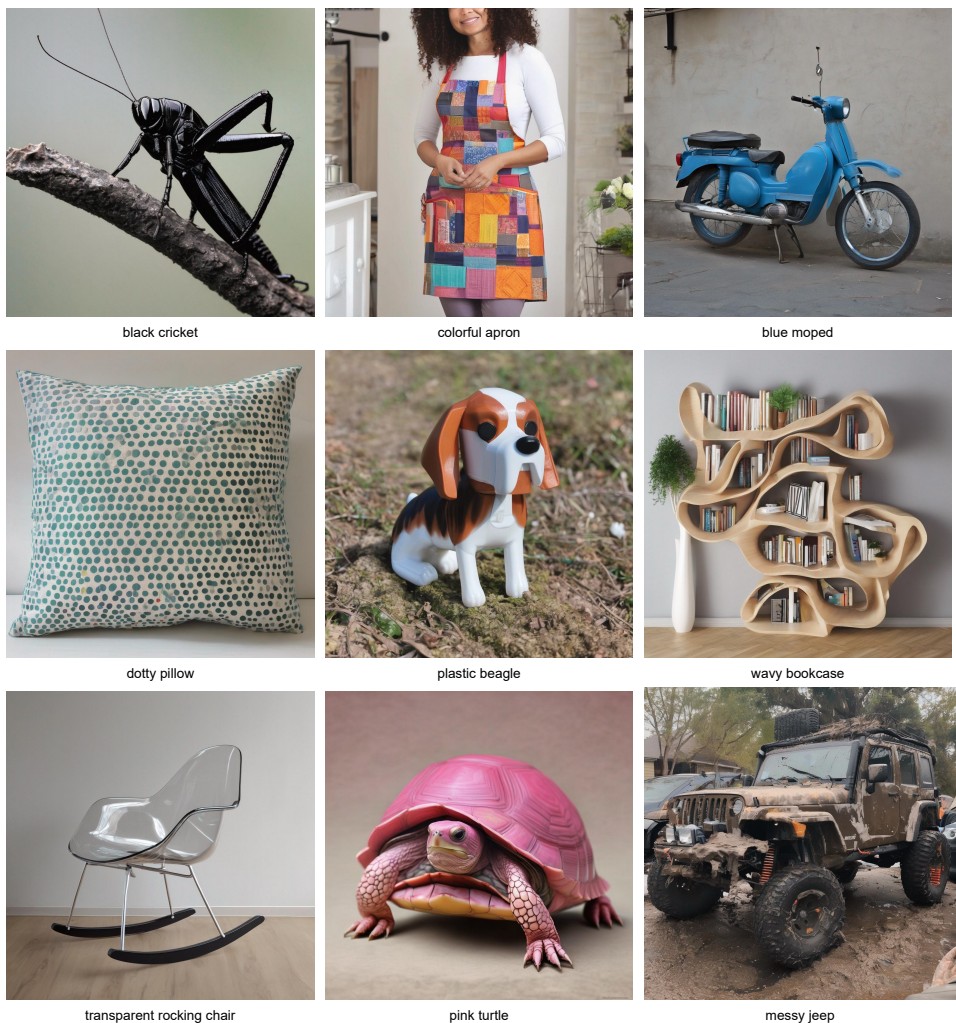

black cricket     colorful apron     blue moped

dotty pillow     plastic beagle     wavy bookcase

transparent rocking chair     pink turtle     messy jeep

Figure 13: Some examples of ImageNet-AO-v2 images generated by SDXL using low-frequency compositions as prompts.

