# OpenReview forum: "CLIP Exhibits Improved Compositional Generalization Through Representation Disentanglement"
_ICLR.cc/2024/Conference — Submitted to ICLR 2024_

### Official Review · Reviewer_PAYB · 2023-10-31

**Soundness:** 3 good
**Presentation:** 3 good
**Contribution:** 2 fair
**Rating:** 6
**Confidence:** 3

**Summary:**

This paper proposes a new dataset to benchmark the compositional capabilities of several CLIP models (OpenAI and OpenCLIP). This dataset is generated using DALLE, and covers the 1000 class names from the ImageNet dataset combined with 30 adjectives. Manual annotators validated the combinations, resulting in ~12k plausible compositions, from which they generated 50k images. The authors also propose to measure the compositional generalization via the normalized mutual information between objects and attributes, and use Z-Diff Score, DCI-Informativeness, Explicitness score, and DCIMIG metrics to evaluate the disentanglement in the embeddings from the models.

**Strengths:**

+ This paper proposes an interesting approach to measure the compositional capabilities of large-scale VL models, by leveraging a text-to-
image model to generate new images with specific attributes.

+ The authors provide a large set of experimental results in the supplementary materials, showing that CLIP models struggle with their proposed dataset

+ This paper is well-structured, easy to read and follow.

**Weaknesses:**

+ There is no description or motivation for the attribute selection, are those attributes randomly selected or generated? How do the authors guarantee that those attributes are not present or co-occur less in the training data?

+ The human validation seems crucial in generating the proposed benchmark; however, there is no detailed description of how this was performed.

+ In section 1, the authors claim: "By assessing the captions in the training sets, we guarantee that none of the captions in our test dataset or similar captions are included in the CLIP training data." -- however, I couldn't find any empirical or theoretical evidence, nor existing reference for this claim.

+ The human validation only asses for the plausibility of the noun-adjective composition, but are the images generated by DALLE following those compositions? Prior work has shown that Diffusion models "struggle to understand the composition of certain concepts, such as confusing the attributes of different objects or relations between objects"[1]. It is unclear if the generated dataset follows the attribute-noun composition, or falls into this category. See also [2].

+ Most of the conclusions are prevalent in the literature (e.g., the diversity of training captions promotes compositionality [3]), and the mutual information analysis does not seem to provide additional insights [4, 5].

[1] Liu, Nan, et al. "Compositional visual generation with composable diffusion models." European Conference on Computer Vision. Cham: Springer Nature Switzerland, 2022.

[2] Park, Dong Huk et al. “Benchmark for Compositional Text-to-Image Synthesis.” NeurIPS Datasets and Benchmarks (2021).

[3] Doveh, Sivan, et al. "Dense and Aligned Captions (DAC) Promote Compositional Reasoning in VL Models." arXiv preprint arXiv:2305.19595 (2023).

[4] Radford, Alec, et al. "Learning transferable visual models from natural language supervision." International conference on machine learning. PMLR, 2021.

[5] Oquab, Maxime, et al. "Dinov2: Learning robust visual features without supervision." arXiv preprint arXiv:2304.07193 (2023).

**Questions:**

Is there any particular reason why DINO-v2 and BEiT-v2 are mentioned briefly in the introduction, but no further analysis is done in the following sections?

---

> ### Author Response · Authors · 2023-11-15
> **Responses to Points 1-4**
>
> Thank you for your valuable feedback. I appreciate the time and effort you have put into reviewing my work. Below, I have addressed each of your points in detail.
>
> **Point 1**: For the selection of attributes, we began with a comprehensive list of adjectives derived from various linguistic sources and large language models such as ChatGPT. This initial list was subjected to a practical evaluation phase where preliminary images were generated and assessed for the visual translation of these adjectives. Attributes that were not clearly represented, such as ’fast’, were discarded. This process resulted in a refined set of 30 adjectives that were visually discernible.
> As mentioned in Section 6.2, we combined these attributes with a list of objects. We then searched for these combinations in the captions of the CLIP training datasets (CC,YFCC,LAION,Datacomp). Combinations found in the training datasets, like "dotty bag", were removed. The remaining combinations, such as "hairy jeep", do not appear in the CLIP training dataset captions. Therefore, these combinations can be considered out-of-distribution data for CLIP models trained on those datasets. It is important to clarify that we do not consider the attributes alone to be out-of-distribution, but rather the full combinations of attributes and objects. For example, "hairy" on its own would not necessarily be out-of-distribution, but the novel combination "hairy jeep" would be.
>
> **Point 2**: As outlined in Section 6.2 and visually depicted in Figure 3, our human evaluation protocol involved a straightforward yet effective method for ensuring the quality and relevance of the generated images. Two evaluators were assigned the task of assessing each image against two critical criteria to determine its suitability for inclusion in the dataset:
>
> -**Object Recognition**: Any image where the central object was not clearly recognizable or was ambiguous was removed.
>
> -**Attribute Visibility**: Images were excluded if the specified attributes were not evidently depicted.
>
> **Point 3**: Our methodology involved an initial generation of 30,000 attribute-object combinations. We then conducted a comprehensive search for these combinations within the captions of various datasets on which CLIP has been trained, such as LAION. This search was instrumental in identifying and eliminating any combinations that already existed within the training data. Even if the individual attribute and object words appeared at a distance in a caption, we eliminated the combination of them from our combinations.
>
> Consequently, the remaining 12,000 combinations, which did not appear in the training data, were used to generate our test dataset. This rigorous process underpins our confidence in asserting that our test dataset comprises truly OOD examples relative to the CLIP training datasets.
>
> **Point 4**: We acknowledge the valid concerns raised about ensuring the integrity of the attribute-noun compositions in our generated images, and the potential issues with diffusion models identified in prior work. As detailed in Section 6.2 and shown in Figure 3, our human evaluation process was designed to directly address these concerns. The main motivation for human evaluation is that text-to-image models currently have limitations in generating coherent compositions. We first evaluated different models like Stable Diffusion variants, Deep Floyd, and DALL-E, and found DALL-E to be the most capable at generating compositions. However, since its accuracy is imperfect for every composition, we added human evaluation to our pipeline to remove low quality
> and ambiguous images, and also the images whose contents are not consistent with the specified attribute-object.

---

> > ### Author Response · Authors · 2023-11-15
> > **Responses to Point 5 and Question**
> >
> > **Point 5**:
> > While prior work has explored the relationship between training data diversity and compositional generalization, our study provides new analysis quantifying *how* diversity in the training data leads to improved disentanglement in model representations, thereby directly enabling better out-of-distribution (OoD) generalization.
> >
> > Specifically, we conducted the first analysis (Section 3.3.1) that examines the impact of caption diversity and quality in training datasets on the mutual information (MI) between object and attribute tokens. We found that lower NMI indicates higher disentanglement between the two, which can improve the compositional generalization.
> >
> > Building on this, we propose a way to measure the disentanglement between objects and attributes in the representation spaces of CLIP text and image encoders (Section 3.3.2). To our knowledge, this disentanglement has not been explored or validated before in CLIP models. We demonstrate models with more disentangled representations achieve higher performance on our new compositional OoD benchmark. We also provided evidence that the disentanglement is much stronger on the text representation, and provided some theoretical insights on how the disentanglement could be induced on the image representation from the text.
> >
> > Furthermore, we provide additional insights into CLIP disentanglement beyond our benchmark by evaluating on 3D Shapes datasets. Our work is the first to extensively examine and validate the connection between disentangled representations and compositional generalization performance of CLIP models.
> >
> > Overall, our analysis and experiments on quantifying disentanglement and its impact on compositional generalization enhance the understanding of why and how CLIP models are able to generalize compositionally. We extend current knowledge by empirically demonstrating that diversity promotes compositionality through inducing disentangled representations.
> >
> > **Question**: Since DINO-v2 and BEiT-v2 are primarily image encoders without an accompanying text encoder, they fall outside the scope of our evaluation criteria. Most of our experimental design required models with text encoding capabilities to comprehensively assess and compare their performance.

---

> ### Comment · Reviewer_PAYB · 2023-11-22
>
> After carefully reading all reviews and responses, I'm keeping my original rating (5) and upgrading my confidence to 5. All reviewers have a consensus regarding some claims in the paper, especially regarding attribute selection and compositionality claims. Moreover, there are several aspects that remain unclear: the lack of information regarding human evaluations along with the confusing narrative (as pointed out by Reviewers unRs and qiWV). I would suggest to the authors to revise and resubmit to another venue.

---

> > ### Author Response · Authors · 2023-11-22
> > **Please checkout our response**
> >
> > Dear reviewer,
> >
> > While we admit some shared concerns between the reviewers at the beginning, we provided several clarifications in our response that might have been overlooked. For instance, we provided the detailed information on how the attributes are selected in the rebuttal, which is NOT further criticized by any of the reviewers. Also, could you please mention which compositionality claims in the paper is unsupported or has issues? We provided several further evidences along the disengagement of representations results in our response.
> > Human evaluation has also been addressed in our response and we believe that we clearly mentioned the criteria that are used to make the assessment (e.g. object recognition and attribute visibility). We kindly ask the reviewer to guide us on which aspect of this assessment is still unclear to him/her.

---

> > ### Comment · Area_Chair_FE77 · 2023-12-02
> > **Re: Official Comment by Reviewer PAYB**
> >
> > Dear Reviewer PAYB:
> >
> > Thanks for the comments on the authors' rebuttal. The authors have posted additional clarification on the focus of their paper and the dataset design details. Please check if you have overlooked any details here:
> > --------
> > 1. Addressing Shared Concerns and Providing Clarifications: While we acknowledge the shared concerns initially raised by the reviewers, we believe that our comprehensive clarifications provided in the response might have been overlooked. Specifically, we detailed how the attributes for our study were selected, which was a key point of concern. Notably, this aspect of our response did not receive further criticism from any of the reviewers, suggesting that our clarification was satisfactory.
> >
> > 3. Emphasizing the Primary Contribution of Our Paper: We would like to reemphasize that the primary focus of our paper is on how increased disentanglement in CLIP representations aids in generalizing compositional Out-of-Distribution scenarios. This is substantiated by experiments on two datasets, including our custom-generated dataset and the 3D shape dataset. The consistent results strongly support our hypothesis, and we believe this critical aspect may have been somewhat overshadowed in the discussions about dataset design.
> >
> > 4.The Authenticity of Our Dataset Design: In order to ensure the validity of our dataset design, we adhered to the following key principles:
> >
> > Comprehensive and Systematic Attribute and Object Selection: We meticulously selected a diverse range of objects (representing entire ImageNet classes) and comprehensive attributes. This approach aimed to ensure inclusivity., and systematic representation.
> >
> > Ensuring Originality: We took care to thoroughly validate the uniqueness of the captions in our dataset, guaranteeing no overlap with any of the captions from current open-source training datasets utilized for CLIP models. Any captions discovered to match those in the training sets were eliminated from our dataset.
> >
> > Comprehensive Human Evaluation: To guarantee the reliability of our dataset, we performed exhaustive human evaluations. Two separate reviewers stringently examined the captions to validate they accurately portrayed the corresponding images. Any instances of disagreement were meticulously re-evaluated to ensure consistency and accuracy across the dataset. Through this comprehensive review process, we upheld precision and fidelity in our data.
> >
> > --------
> >
> > Please review their response and adjust you scores if the response address your concerns.
> >
> > AC

---

> ### Comment · Reviewer_PAYB · 2023-12-04
>
> Dear AC,
>
> After carefully reviewing the comments and discussions, I'm keeping my score. There is no real clarification with respect to the human evaluations, and some claims need revision (e.g., Any captions discovered to match those in the training sets were eliminated from our dataset). This is only true for OpenCLIP, not all CLIP models, which sustain the author's claims with respect to the conducted evaluations.
>
> Furthermore, I don't see any revision based on the reviewer's and authors' discussion. There is no revision to the supplementary material, and it's unclear what portion of all major points raised in the discussion period will go in the main paper or appendix (e.g. The full methodology for selecting attributes will be detailed in the Appendix -- however this is still unclear)

---

> > ### Comment · Area_Chair_FE77 · 2023-12-05
> >
> > Dear Reviewer PAYB:
> >
> > The authors have the following comments to address your concerns. Please let me know if they addressed your concerns.
> > Thanks,
> >
> > AC
> >
> > ---------------Comments from the authors:
> >
> > #Clarification on CLIP Models and Datasets:
> >
> > You mentioned that the claim regarding the elimination of captions from training sets applies only to OpenCLIP and not all CLIP models. It's important to clarify that this approach was indeed applied to all models evaluated in our study. We thoroughly investigated and discovered datasets for each model, including CC12m, YFCC15m, LAION, and Datacomp. The only exception was the OpenAI model, which uses a private dataset.
> > It's essential to note that OpenCLIP is not a singular model but a comprehensive project and repository that amalgamates various available CLIP models. OpenCLIP stands out as one of the most popular repositories on GitHub within the field of CLIP and VLM models. Many research papers and teams dedicated to CLIP utilize this repository as a central hub, contributing various CLIP models, and the latest advancements are consistently added to enrich this collaborative resource. Repository link : https://github.com/mlfoundations/open_clip
> >
> > #Revisions and Supplementary Material:
> >
> > Regarding the lack of revisions and updates to supplementary materials, I assure you that we have made substantive revisions based on the discussions between reviewers and authors. These revisions will be more apparent in the final version of the paper. Please see the next comment, where we outlined the items to be added to the Appendix.
> > We understand the importance of detailing our methodology for selecting attributes and ensuring transparency. Therefore, we have included a comprehensive explanation in the Appendix of the final paper. This addition aims to address the major points raised during the discussion period and provide clarity on our methods. Please see the next comment for further details.

---

### Official Review · Reviewer_qiWV · 2023-11-01

**Soundness:** 2 fair
**Presentation:** 2 fair
**Contribution:** 3 good
**Rating:** 5
**Confidence:** 3

**Summary:**

This paper studies CLIP models under a different type of distribution shift namely compositional OOD generalization, where the objects and attributes may be individually seen during training, but their composition is unseen. A new dataset, ImageNet-AO is generated using DALL-E, containing such novel compositions for ImageNet classes. It is ensured that the generated compositions are not present in the CLIP training datasets. Key observations are - i) compositional diversity of the training dataset improves the compositional generalization of the CLIP model, ii) image/text representation disentanglement of objects and attributes improves generalization, iii) larger, more diverse datasets leads to better compositional generalization, iv) better disentanglement in representations leads to better compositional generalization.

**Strengths:**

- The experiments are well-designed
- Conclusions drawn are very interesting and insightful
- The dataset ImageNet-AO can be helpful for future study as well

**Weaknesses:**

- "*the training dataset of the OpenAI CLIP has not been released, which makes designing a test set that has a truly different distribution from the training one challenging. We aim to address these issues in this paper, by focusing our attention on the compositional generalization
in the single object setting, and designing an authentic test dataset to assess the training data characteristics and mechanisms in the models that lead to the OoD generalization.*" -- This contradicts the following statement where the authors claim that they verify that the ImageNet-AO images are not a part of several CLIP training dataset -- "*To ensure these combinations were not present in the CLIP training set, we conducted a thorough search and removed any combinations that were found.*"
- "*By assessing the captions in the training sets, we guarantee that none of the captions in our test dataset or similar captions are included in the CLIP training data.*"
    - Is this check done for all the other datasets considered in the paper as well (LAION, YFCC15m, CC12m, and DataComp)?
    -  A similar check should be done on images as well, it is possible that such images with different captions are present in the training set. Usually, captions from web sources are not exactly descriptive of the image.
- "*We also found that the CLIPs that show higher OoD generalization typically exhibit strong disentangled text representations. Furthermore, such CLIPs also enjoy a more disentangled image representation with respect to the attributes and objects as well.*" -- the experiments in the paper do hint at the above statement. But this does not necessarily imply the following: "*Specifically, a dataset with diverse compositions of attribute-objects facilitates a more disentangled text representation, which in turn induces a disentangled image representation through contrastive learning.*" It could be possible that diverse images lead to disentangled image representations as well.
- "*To evaluate the degree of disentanglement in the training captions utilized by the CLIP, we conducted an analysis by measuring the normalized mutual information (NMI) between the object class and attribute tokens, whose domains are defined based on the captions in our generated dataset.*" -- Could the authors explain how the domains are defined based on the captions in the generated dataset? More details on how the NMI is measured would be helpful.
- Fig.4 - It is not clear how the disentanglement metrics are computed for the image encoder.
- "*We aimed for a diverse set of class names to enhance the complexity of the generated images.*" -- It is not clear if all 1000 classes were used or only a subset. If a subset was used, how was this chosen?
- "*This dataset was produced by creating compositional images via a text-to-image model, using an Attribute+Object template.*" -- could the authors give more details/ a citation for the Attribute+Object template?
- Could the authors provide details on where the 30 adjectives were chosen from?
- "*Lastly, human evaluation was used to validate the generated images, with images not closely aligning with their prompts removed. After this process, around 12000 combinations remained, for which we successfully generated near 50000 accurate, high-quality images.*" - The order of the two statements may have to be swapped? Could the authors provide details on how this human evaluation was done?
- "*For the test sets, all 1000 classes of ImageNet were used as the in-distribution set and expanded the number of classes to approximately 12000 for the OoD set.*" -- could the authors share how the captions were created for the OOD set? Sharing some examples would be helpful. I believe the 80 captions are used only for the ID set, and single relevant captions are used for the OOD set?
- In Fig.1, for a more fair comparison, the image-only models such as DINO-v2 and BEiT-v2 should also be trained on the datasets that were used for training CLIP (by using only the images, and ignoring the captions). Without matching at least the image datasets, there is not enough evidence to support the following statement - "*We interpret these findings as strong evidence that the inclusion of language supervision, particularly during CLIP training, positively impacts the model representation quality, hence making it possible to generalize to unseen compositions, despite the absence of such compositions in their training data.*"

Nitpicks -

- citation format seems non-standard - (x) vs. [x]
- inline citations should use the format xyz et al., rather than [x]
- A citation for the work that defines "compositional OOD generalization" would be helpful

**Questions:**

- Although the experiments and conclusions in the paper are interesting and useful, several aspects of the paper need more clarity. These are mentioned in the weaknesses section. I will be happy to update my score based on clarifications provided by the authors.
- Codes, models, and datasets must be open-sourced for the benefit of future research. Could the authors comment on this? Would these be released upon acceptance?

---

> ### Author Response · Authors · 2023-11-15
> **Answer Point 1, 2 and 3**
>
> Thank you for your valuable feedback. I appreciate the time and effort you have put into reviewing my work. Below, I have addressed each of your points in detail.
>
> **Point 1**: It is true that the specific training dataset used by OpenAI for CLIP has not been publicly re-
> leased. However, there are several multimodal datasets that have been made available and are commonly
> used for training vision-language models, such as LAION, YFCC, CC, and DataComp. Our work utilizes
> these publicly available datasets to construct a test as mentioned in Section 6.2. We also evaluate CLIP
> models trained on these public training datasets.
>
> **Point 2**: We have thoroughly checked all datasets referenced in our paper, including LAION, YFCC15m,
> CC12m, and DataComp, to ensure that (attribute,object) pairs in our dataset are not present in the
> captions of these CLIPs’ training data.
> Since we want to evaluate models on novel compositions, we first found (attribute, object) pairs that
> are not appear in any captions of the available training sets of CLIPs. For these novel (or extremely rare)
> combinations, the images were also generated which makes the possibility of image overlap extremely
> unlikely. For instance, we have combinations such as ‘hairy jeep,’ or ‘luminous golf cart,’ in our dataset,
> as illustrated in Fig. 2, which are unlikely to be encountered on the web.
> While we acknowledge the possibility that the model may have been exposed to similar images during
> training, we can assert with confidence that the exact image and complete caption pairs used in our
> test dataset are unique and represent new combinations that the model has not previously encountered.
> This pairing approach ensures that, despite any potential familiarity with individual images, the model
> is still being evaluated on its ability to generalize to new image-caption combinations, which we have
> engineered to be out-of-distribution (OoD). Additionally, the principal experiments presented in our
> paper, particularly in Section 3.1, as well as the supplementary experiments detailed in Sections 4.2, 6.3,
> and 6.4, incorporate both image and text components within their methodologies. These experiments
> do not solely concentrate on the image aspect.
>
> **Point 3**: The textual input is inherently disentangled, with adjective and object tokens being *distinctly separate*. In contrast, the visual inputs inherently exhibit entanglement; attributes and objects within images are naturally intertwined. By augmenting the diversity of textual data, we promote disentanglement in text representation, as the model learns to distinguish between the separate tokens for adjectives and objects due to their consistent and isolated presentation. However the entanglement in image representations persists because the attributes and objects are not presented as isolated features but as parts of an integrated whole.  Therefore, while increased diversity can enhance the model's exposure to varied visual concepts, it does not automatically lead to disentangled representations. These claims are also backed by Fig. 4, where we see that the text encoder usually exhibits a more pronounced disentanglement score compared with the image encoder.
>
> In the table below, we also compare a CLIP model with some image-only models that were trained on large-scale datasets or are among the top-performing models on ImageNet benchmarks. We evaluate these models on different disentanglement metrics  on the ImageNet-AO.
>
> **Table: Disentanglement metrics on different Models**
>
> | Model                     | z_diff ↑ | DCI-I ↑ | explicitness ↑ | DCIMIG ↑ |
> |---------------------------|----------|---------|----------------|----------|
> | CLIP-ViT-14-datacomp-xl   | 0.955    | 0.1834  | 0.9103         | 0.025    |
> | DINOv2                    | 0.95     | 0.1787  | 0.9022         | 0.021    |
> | Beit                      | 0.901    | 0.1463  | 0.884          | 0.016    |
> | resnext101      | 0.948    | 0.1745  | 0.901          | 0.0209   |
> | Effeicent-b7              | 0.899    | 0.1274  | 0.8531         | 0.0134   |

---

> > ### Author Response · Authors · 2023-11-15
> > **Responses to Points 4-9**
> >
> > **Point 4**:
> >  In the context of Normalized Mutual Information (NMI) in our experiments, 'X' represents the entire set of attributes that we utilized in generating our dataset, encompassing various descriptive elements such as texture. 'Y', on the other hand, corresponds to the set of objects used in generating our dataset, including a range of specific items or entities. These definitions of 'X' and 'Y' enable us to analyze the relationship between attributes and objects in our dataset as learned by the CLIP model.
> >
> > $$
> > \\begin{aligned}
> > & \\mathrm{MI}(\\mathrm{X} ; \\mathrm{Y})=E_y[\\mathrm{H}(\\mathrm{X})-\\mathrm{H}(\\mathrm{X} \\mid \\mathrm{Y}=\\mathrm{y})] \\\\
> > & \\mathrm{H}(\\mathrm{X})=-\\sum_{i=1}^n p_i^* \\log \\left(p_i\\right) \\\\
> > & p\\left(x_i\\right)=\\frac{\\# \\text { captions that have } x_i}{\\sum_{j=1}^n \\text { \\#captions that have } x_j} \\\\
> > & p(x, y)=\\frac{\\# \\text { captions that have } x \\text { and } y}{\\sum_{x y} \\sum_y \\text { captions that have } x_i \\text { and } y_j} \\\\
> > & \\mathrm{H}(\\mathrm{X} \\mid \\mathrm{Y}=\\mathrm{y})=-\\sum_x p(x, y) * \\log \\left(\\frac{p(x, y)}{p(y)}\\right) \\\\
> > & \\mathrm{NMI}(\\mathrm{X} ; \\mathrm{Y})=\\frac{2 * M I(X ; Y)}{[H(X)+H(Y)]}
> > \\end{aligned}
> > $$
> >
> > **Point 5**: As mentioned in Section 3.3.2, we can consider two super latent factors corresponding to attributes and objects. For each vector of factors, we choose a image associated with those factors. We then calculate the embedding for that image. This gives us a set of representations and corresponding latent factors. Using these embeddings and factors, we can calculate different disentanglement metrics. To assess the randomness effects, we repeated this experiment multiple times,  and observed that the results show negligible variance. We also observed that if instead of random selection, we calculate embeddings of all images per relation and take the average, the results hold. We note that this is because the images associated with a certain relation have similar embeddings. For more details on how each metric is calculated, please refer to [2].
> >
> > **Point 6**: As detailed in Section 6.2 of the Appendix, we utilized the entire spectrum of 1000 classes from
> > the ImageNet dataset.
> >
> > **Point 7**:  We appreciate the opportunity to provide additional details on the methodology used to generate our dataset. As mentioned in Section 6.2 and illustrated in Figure 3, we employed a simple template structure, [ATTRIBUTE] [OBJECT], to create compositional images via a text-to-image model. This approach allowed us to systematically combine various attributes with objects to enrich the diversity and complexity of the images. Also this approach was used in previous works [3,4]. For concrete examples of these prompts, one may refer to Figure 2 in the main text, which showcases sample image captions generated using this template. Furthermore, a more extensive collection of prompts can be found in Figure 8 of the Appendix.
> >
> > **Point 8**: Initially, we compiled an extensive list of adjectives using various sources, including linguistic websites and large language models like ChatGPT, to ensure a broad and inclusive initial selection. The primary list contained 50 attributes. Through iterative experimentation, we generated preliminary images for a range of objects paired with these adjectives. This practical evaluation was crucial as it allowed us to observe the representation of certain attributes within the images. We found that some adjectives, such as ‘fast’, did not visually translate well into the generated images. Consequently, these were omitted from the final list. The refinement process led us to a curated set of 30 adjectives that were visually discernible and thus suitable for our image generation objective. The full methodology for selecting attributes will be detailed in the Appendix.
> >
> > **Point 9**: We have to clarify that by 'after this process ...,' we meant to report the number of generated images after the final step, which is the human inspection. Here we got total of 50k images from 12k combinations that passed the human evaluation criteria. The human inspection was conducted in two stages: First, before generating any images, we filtered the initial list of 30,000 combinations by searching for these combinations in the captions of training datasets and removing any matches, to ensure the remaining combinations were novel and feasible to depict visually. Second, after generating images for the remaining novel combinations using the text-to-image model, further human evaluation was necessary to eliminate unsatisfactory images where either the object was not clearly rendered or the intended attribute was not distinctly perceptible. Through this two-step refinement process, we narrowed down the dataset from 30,000 initial combinations to a final curated collection of around 12,000 novel  (attribute,object) combinations comprising 50,000 images that passed the human evaluation criteria.

---

> ### Author Response · Authors · 2023-11-15
> **Responses to Points 10,11**
>
> **Point 10**: Our dataset contains images generated from textual prompts, which are then used as captions for evaluation. For example, the "hairy jeep" image was generated from the prompt "hairy jeep".
>
> In zero-shot evaluation, ID image embeddings are matched against 1,000 captions for the 1,000 classes. OoD image embeddings are matched against 12,000 captions since each combination is a unique class.
>
> For both sets, class name captions are put into 80 templates like "this is a photo of..." to create multiple embeddings. The final class embedding is the average of these 80 variations.
>
> **Point 11**: As delineated in Section 3.2, the aim of our study was not to forge a direct and fair comparison between CLIP and image-only models like DINO-v2 and BEiT-v2. Instead, our focus was investigating the mechanism that causes the language supervision to improve  CLIP's out-of-distribution (OoD) generalization capabilities as mentioned in the title and abstract of our paper. We selected the supervised and self-supervised models for comparison based on their state-of-the-art ImageNet performance and their training on large-scale datasets, which allowed us to make a closer, albeit approximate, comparison given our practical constraints.
> Only to see performance of other powerful foundation models (that are not based on language supervision), we show the results of these models on our proposed dataset too.
>
>
> The evidence supporting the role of language supervision in enhancing CLIP's performance is multifaceted, including:
> - Variations in performance among models trained on Conceptual Captions (CC) and YFCC datasets, particularly with lower Normalized Mutual Information (NMI).
> - A clear disentanglement at the representation level.
> - A well-documented correlation between representation disentanglement and OoD generalization.
>
> Our narrative underscores the distinctive benefits of language supervision and its impact on OoD accuracy, rather than on a direct performance comparison. Given the logistical challenges of compiling a dataset comparable to the one used to train CLIP, we opted to showcase the most relevant models to support our hypothesis.
>
> At the end, about your question, we plan to open-source all codes and datasets upon the paper’s acceptance to
> support future research.
>
> [3] Lewis, Martha, et al. "Does clip bind concepts? probing compositionality in large image models." arXiv preprint arXiv:2212.10537 (2022).
> [4] Nayak, Nihal V., Peilin Yu, and Stephen H. Bach. "Learning to compose soft prompts for compositional zero-shot learning." arXiv preprint arXiv:2204.03574 (2022).

---

> > ### Comment · Reviewer_qiWV · 2023-11-22
> >
> > I thank the authors for the detailed and thorough response, which addresses several of my concerns. Please find my comments below-
> > - **Point 1**: In this case, the following statement needs to be removed from the Introduction - "For instance, the training dataset of the OpenAI CLIP has not been released, which makes designing a test set that has a truly different distribution from the training one challenging. We aim to address these issues in this paper, by ... "
> > This is because, the construction of ImageNet-AO assumes access to the training dataset of  CLIP, while this statement makes it seem like access is not needed.
> > - **Point 2**: It is possible that synonyms of object-attribute pairs exist as captions to similar images. Although the exact pair may not exist in the training dataset, a more robust means of ensuring that a similar image-caption combination does not exist in the training dataset is required. For example, finding the k nearest neighbors of the caption and image individually, and using human inspection to ensure the k nearest captions and k nearest images are not similar to them. Since the training dataset is very large scale, we cannot rely on the combination being unlikely to exist in the dataset, although it seems to be the case.
> > - **Point 3**: Could the authors clarify whether the first row in the table presented in the rebuttal uses text encoder or image encoder? Also, could the authors share the conclusions from the table?
> > - **Point 9**: For human evaluation, it is important to also include details on how this was conducted, and how many people participated in the study.
> >
> > I update my score to 5 based on the rebuttal.

---

> > > ### Author Response · Authors · 2023-11-22
> > > **Responses to Points 1,2,3**
> > >
> > > Thank you for sharing your insightful feedback. In the following response, I've thoughtfully addressed each of your points.
> > >
> > > **Point 1**:
> > > In making a dataset whose probability of overlap with that of the OpenAI CLIP training set, we aimed to make images with novel compositions that are not observed frequently in the real-world. While we do not have access to the OpenAI CLIP training captions, we took captions in other training sets that are used in other versions of CLIPs as a surrogate for real-world captions. By generating images based on prompts that are extremely rare in the real world, we make the chance of overlap between our data and that of the OpenAI CLIP small. Hence, in making our dataset by this strategy we do not actually need to have access to the OpenAI training captions.
> > >
> > > Also, in contrast to previous studies, such as [1], where different CLIP models are evaluated in Out-of-Distribution contexts using datasets like ImageNet v2 or ImageNet R, it is important to note that there is no definitive evidence confirming these datasets were excluded from training sets like LAION. However, our dataset provides this evidence.
> > >
> > > [1]Fang, Alex, et al. "Data determines distributional robustness in contrastive language image pre-training (clip)." International Conference on Machine Learning. PMLR, 2022.
> > >
> > > **Point 2**:
> > > We appreciate your suggestion regarding the verification of the uniqueness of object-attribute pairs in our dataset, especially in the context of the large-scale training dataset. We recognize the importance of ensuring that not only the exact pairs but also similar combinations are not present in the training dataset.
> > > In our current study, we prioritized generating rare combinations of attributes and objects, focusing on creating unique pairs for zero-shot evaluations. This approach was aimed at minimizing the likelihood of overlap with existing datasets, given the constraints we faced in terms of resources. While we have endeavored to observe and maximize the difference in distribution compared to existing datasets that claim to OoD, we acknowledge that there is always room to enhance this aspect further.
> > > We acknowledge the merit of your suggestion to use methods like identifying the k nearest neighbors for both captions and images. This could indeed provide a more robust assurance of the uniqueness of our dataset and help to extremize the distinction from other datasets. If we can secure sufficient resources in the future, we plan to incorporate these more rigorous checks into our methodology. This would not only align with our goal of creating a distinct dataset but also significantly enhance the reliability and validity of our evaluations.
> > >
> > > **Point 3**:
> > > To clarify, the first row in the table indeed uses the image encoder. This is evident from our juxtaposition of the CLIP model alongside image-only models like DINOv2 in the same table. It's important to recognize that DINOv2 is acknowledged as a state-of-the-art model in various tasks, as detailed in their paper[2].
> > > The key takeaway from the table is the superior performance of the CLIP model in terms of disentanglement metrics. Despite CLIP and models like DINOv2 being trained on millions of images, CLIP demonstrates a higher level of disentanglement in its representations.
> > > Additionally, it's worth noting that the level of disentanglement in the text encoder of the CLIP model is even higher than that in the image encoder(figure 4). This observation aligns with our discussion in section 3.3.3, where we delve into how the diversity of the training dataset enhances the disentanglement capabilities of the text encoder. This improved disentanglement in text representation subsequently contributes to the enhanced disentanglement in image representation.
> > > Furthermore, our findings suggest that training solely on a diverse image dataset does not inherently lead to highly disentangled representations. This aspect becomes particularly evident when considering the model's accuracy in OoD scenarios, as demonstrated in figure 1-right of our paper.
> > >
> > > [2]. Oquab, Maxime, et al. "Dinov2: Learning robust visual features without supervision." arXiv preprint arXiv:2304.07193 (2023).

---

> > > > ### Author Response · Authors · 2023-11-22
> > > > **Response to Point 9**
> > > >
> > > > **Point 9**:
> > > >
> > > >  We acknowledge the reviewer's concern regarding the lack of detailed information on the human evaluation process in our paper. We will address this oversight and provide a comprehensive description in the revised manuscript. While we briefly mention this aspect in section 6.2, we agree that a more detailed account is necessary.
> > > > In the human evaluation phase of our study, we engaged two evaluators for the assessment of images. Their task was to ensure the quality and relevance of each image according to two primary criteria:
> > > >
> > > > **1. Object Recognition**: The evaluators scrutinized each image to confirm that the central object was clearly recognizable and unambiguous. Any image where the object could not be easily identified or was subject to interpretation was excluded from the dataset.
> > > >
> > > > **2. Attribute Visibility**: The evaluators also assessed whether the specified attributes were visibly and unmistakably depicted in the image. Images that failed to clearly exhibit the required attributes were removed from consideration.
> > > > The process involved a systematic review of each image by the evaluators. Each image was independently assessed by both evaluators, and any discrepancies in their evaluations were resolved through discussion to reach a consensus.
> > > > We will include these details in section 6.2 of our revised paper to provide a clear understanding of our human evaluation process and its role in enhancing the quality of our dataset.

---

> > > > ### Author Response · Authors · 2023-11-23
> > > >
> > > > Thanks again! We would appreciate if the reviewer could kindly take our response to his 3 points that have just been raised into consideration. We are willing to make further clarification and experimentation per the reviewer’s request.

---

> > > > > ### Comment · Reviewer_qiWV · 2023-11-23
> > > > >
> > > > > I thank the authors for their prompt response. I have no further questions for the authors.
> > > > >
> > > > > Although the experiments and insights in the paper are useful, I maintain my rating of 5 since the dataset presented requires a more robust human evaluation to ensure that it meets the criteria claimed in the paper. Further, I believe that a more formal human evaluation is important for releasing a dataset. I am open to revising my rating based on discussions with other reviewers/ AC.

---

> > > > > > ### Author Response · Authors · 2023-11-23
> > > > > > **Response to Reviewer for Clarification on Data Generation**
> > > > > >
> > > > > > To provide more clarity on the human evaluation aspect, we have compiled statistics related to the data generation and human evaluation in the following table:
> > > > > >
> > > > > > | Image Generation Count | Images Remaining After Evaluation | Images Removed | Percentage of Removal Due to Missing Object | Percentage of Removal Due to Attribute Failure |
> > > > > > |------------------------|----------------------------|----------------|---------------------------------------------|-----------------------------------------------|
> > > > > > | 74,500                 | 50,100                     | 24,400         | 10%                                         | 90%                                            |
> > > > > >
> > > > > > From the table, it is evident that the majority of image removals are attributed to issues with attribute binding, a known limitation of text-to-image (T2I) models. This challenge, particularly the model's struggle to accurately bind attributes to objects, has been discussed in previous works such as Feng et al. [1].
> > > > > >
> > > > > > Additionally, this paper does not aim to highlight a method influenced by dataset bias. Instead, our focus is on evaluating different CLIP models trained on varied datasets to understand the impact of training data on out-of-distribution generalization. The performance trends observed in these models on our dataset are consistent with findings reported in other studies [2, 3, 4].
> > > > > >
> > > > > > References:
> > > > > > 1. Feng, Weixi, et al. "Training-free structured diffusion guidance for compositional text-to-image synthesis." arXiv preprint arXiv:2212.05032 (2022).
> > > > > > 2. Cherti, Mehdi, et al. "Reproducible scaling laws for contrastive language-image learning." Proceedings of the IEEE/CVF Conference on Computer Vision and Pattern Recognition. 2023.
> > > > > > 3. Nguyen, Thao, et al. "Quality not quantity: On the interaction between dataset design and robustness of clip." Advances in Neural Information Processing Systems 35 (2022): 21455-21469.
> > > > > > 4. Ilharco, G., et al. "OpenCLIP (Version 0.1) [Software]." https://github.com/mlfoundations/open_clip.

---

### Official Review · Reviewer_unRs · 2023-11-05

**Soundness:** 3 good
**Presentation:** 2 fair
**Contribution:** 3 good
**Rating:** 6
**Confidence:** 3

**Summary:**

In this work, author examine the compositionally generalization in vision language model. By adopting different combination of disentangled attribute in training dataset of CLIP, author generate a authentic test set that is unseen by model but share the same disentangled attribute. Author also argue that the level of feature disentanglement is high correlate to model generalization by presenting various analysis.

**Strengths:**

1. This paper propose a high quality test set measuring compositional generalization with generative model. This benchmark provide a simply and more straightforward measurement for compositional generalization of Top1 accuracy for synthetic dataset, over prior measurement like using Visual Genome, or captions perturbation. This could be significant to the community exploring model generalization.
2. Author have conducted various analysis over the relationship between compositionally and feature disentanglement, demonstrate the potential influence of the proposed dataset at a large scale.

**Weaknesses:**

1. In 3.2, the statement 'We interpret these findings as strong evidence that the inclusion of language supervision, particularly during CLIP training, positively impacts the model representation quality' might be too strong of a claim. As explored in prior work("Data Determines Distributional Robustness in Contrastive Language Image Pre-training (CLIP)") , language supervision might not be the sole reason for model generalization. There're multiple variance between VLM and other modality and author should not attribute such improvement solely on language supervision.
2. Conclusion are less convincing due to the limited candidate in each experiment. For instance, in Table 1, it will be interesting to shows the NMI for a subset of LAION with the same number of data to other dataset. Also in table 2, there's only 4 results, please consider adding more variance of dataset and CLIP architecture .
3. The narrative after section 4 is a bit too rush, it's hard to follow the method and results. For instance, what is 'dimensions' in 4.1 stands for? And more context over 'switching dimension' would be helpful. Moreover, I cannot tell how the conclusion of 'A higher level of accuracy in the image retrieval task indicates that the model embeddings are more disentangled.' can be drawn from experiment in 4.2.
4. There's some grammar and formatting issue, for instance in section 4, spaces were missing between sentences.

**Questions:**

Page 3: in the imrpoved generalization -> typo
Please refer to weakness. While this work could be potential significant to the community, the clarity could be further improve, especially on drawing the connection between compositionally and feature disentanglement.

---

> ### Author Response · Authors · 2023-11-15
> **Answer to Point 1**
>
> Thank you for your valuable feedback. I appreciate the time and effort you have put into reviewing my work. Below, I have addressed each of your points in detail.
>
> **Point1** : We acknowledge that, as already mentioned in the paper (first paragraph in the introduction), some prior work such as [1] suggested that the model generalization is indeed influenced by multiple factors beyond language supervision alone.
>  However, we note that there are certain shortcomings in assessing the role of language in generalization of CLIP models in [1]. Specifically, the dataset (ImageNet-Captions) that is used in [1] to investigate the effect of language in domain robustness could lack domain information (e.g. painting, sketch, clip art, etc.) in the captions that are used for training and thus does not utilize the language capability to tackle domain shift. This stems from the fact that in this dataset, the captions are taken from the original text data of the Flickr images. Hence, the language is not used in its maximum capacity to train the model. However, in our case, we expect all the composition constituents, which are attributes and objects, been described numerous times *individually* in the training data. In fact, we investigate compositional OoD generalization, which assumes that the novel combination of *known* concepts appear at the test time and evaluate the role of language in this type of OoD generalization. Aside from this, ImageNet-Captions is orders of magnitude smaller (~460k images) than the datasets that are used to train CLIPs, which are often in the order of 100 millions or few billions of images, which could be insufficient to expect the language to play a significant role in enhancing the generalization. On the other hand, we benchmark CLIPs that are trained with much larger training sets. Therefore, our work makes a better setup to assess the role of language in the generalization.
>  Therefore, one of our aims was to re-assess whether the language could bring any additional value to the table beyond what has already been known in the  literature.
>
> We have to emphasize that our intent was not to attribute the observed improvement in model performance solely to language supervision, but rather to see if the language could be *one of the* the  drivers of improvement in our compositional generalization setting. To make this happen, we conducted *controlled* experiments  to specifically isolate and understand the impact of language factors on model generalization. For instance, we compared models with *identical structures* and *training methodologies*, with the only variable being their *training datasets*, such as the CC and YFCC models. Both models employ the same image and text encoders, yet they yield different outcomes. Further analysis revealed that the captions in the CC dataset are of markedly higher quality (table 1), which provides empirical evidence of the language factor's influence on model performance.
>
> We believe these controlled comparisons are essential to discern the discrete contributions of different variables to model generalization. In future iterations of our manuscript, we will strive to more precisely articulate the multifaceted nature of model generalization, ensuring that the significance of language supervision is presented as one of several contributing factors, rather than the sole determinant.
>
> We hope this clarification addresses your concerns, and we are committed to revising our manuscript to reflect a more nuanced understanding of the elements that contribute to the robustness of model generalization.
>
> [1] Fang, Alex, et al. "Data determines distributional robustness in contrastive language image pre-training (clip)." International Conference on Machine Learning. PMLR, 2022.

---

> > ### Author Response · Authors · 2023-11-15
> > **Answer to Point 2 and 3**
> >
> > **Point2** : Table 1 in Section 3.3.1 now includes NMI values for various subsets of the LAION dataset. It is worth noting that these different LAION subsets originate from the same source but undergo different filtering processes. The filtering appears to alter the token distribution, which in turn affects the NMI values. However, NMI has no direct relationship to the dataset size. For instance, as the results indicate, despite its smaller size, LAION 400m shows a bit higher composition diversity compared to that the LAION 2B.
> >
> > **Table1: Normalized Mutual Information between the attributes and objects calculated for captions of some CLIP training sets. The domain of these random variables are defined based on the compositions present in our generated dataset.**
> >
> > | **Dataset** | **Dataset Size** | **NMI**  |
> > |-------------|------------------|----------|
> > | YFCC        | 15m              | 0.9390   |
> > | CC          | 12m              | 0.8903   |
> > | LAION       | 400m             | 0.8307   |
> > | LAION       | 12m              | 0.9024   |
> > | LAION       | 2B               | 0.8541   |
> >
> > Additionally, Section 4.1 now contains expanded results, including additional models in the updated version of Table 2.
> >
> > **Table2: Number of common dimensions across factors and switching dimensions for color manipulation in 3D Shapes dataset**
> >
> > | **Dataset** | **Architecture** | **OoD Accuracy** | **# of Common Dims** | **# of Switching Dims** |
> > |-------------|------------------|------------------|----------------------|-------------------------|
> > | LAION       | ViT-L/14         | 43.14%           | 2                    | 40                      |
> > | LAION       | ViT-B/16         | 38.89%           | 5                    | 60                      |
> > | LAION       | ViT-B/32         | 32.84%           | 7                    | 90                      |
> > | OpenAI      | ViT-L/14         | 39.91%           | 3                    | 5                       |
> > | OpenAI      | ViT-B/16         | 37.73%           | 4                    | 10                      |
> > | OpenAI      | ViT-B/32         | 33.43%           | 6                    | 30                      |
> > | CC          | RN50             | 9.64%            | 15                   | 200                     |
> > | YFCC        | RN50             | 3.60%            | 21                   | 250                     |
> >
> > **Point 3**: In Section 4.1, when we refer to 'dimensions,' we are speaking of the individual elements within the embedding vector generated by the CLIP model. To clarify the concept of 'switching dimensions,' we conducted experiments where these individual elements are systematically altered to assess their impact on the performance of the model. By switching, we mean replacing the values in these dimensions by those of the samples that have a different level of attributes.   This process helps us understand which dimensions of the embedding vector correspond to specific attributes in the data.
> >
> > Regarding the image retrieval task discussed in Section 4.2, we propose that a more decomposable and disentangled representation leads to better retrieval performance. This is because when the embedding of an image is decomposable, adding the embedding of an attribute (like a color or shape) to the image’s embedding results in an alteration of only the intended attribute in the resultant image, without affecting other characteristics. Conversely, if the embedding is entangled, the same operation would inadvertently alter the entire image, not just the intended attribute. Therefore, the ability to perform precise modifications through embedding manipulation is indicative of a disentangled representation, which, in turn, suggests higher accuracy in retrieval tasks.
> >
> > In our additional experiment using the 3D Shapes dataset, we aim to correlate the model's disentanglement metrics with its Out-of-Distribution (OoD) generalization performance. We employ a classifier trained to recognize common features across different images, such as color. By identifying which dimensions of the embedding vector are significant for such features, we gain insights into the disentanglement of the embeddings. Our findings suggest that in higher-performing CLIP models on OoD benchmarks, fewer dimensions need to be switched to achieve an embedding that aligns with a modified attribute, thus reinforcing the connection between disentanglement and OoD performance.
> >
> > We have included additional diagrams in the revised manuscript to visually demonstrate the processes and outcomes of these experiments, specifically experiment 4.1, for greater clarity.
> >
> > We hope that this explanation resolves the ambiguities and look forward to further comments and suggestions.

---

> > > ### Comment · Reviewer_unRs · 2023-11-22
> > > **Update score to 6.**
> > >
> > > Thanks author for the detailed reply. I appreciate update on point 1 and 3, and expect author to revise your draft carefully for more precisely articulation. Besides, I see a nice trend based on including additional experiment for point2, author should also update OpenAI and LAION with more model architecture(like ResNet50-CLIP) to make your experiment more comprehensive.
> > > I will update my score to 6.
> > >
> > > In general I think the idea proposed by this paper is still interesting and insightful, and constructing a dataset based on compositionality is novel and potential impactful to the community. However, as also mentioned by other reviewer, there still many lack of clarity in explaining motivation and justification of the proposed dataset. Author should continue to work on this for future revision.

---

> > > > ### Author Response · Authors · 2023-11-23
> > > >
> > > > Thanks for your insightful comments. We will make sure to revise the paper accordingly, and include more backbones in the results to make the trend clearer. We have to mention that the main motivation behind designing this dataset is to minimize the possibility of train/test overlap in the CLIP models and ensure an authentic validation of OoD generalization of such models. Previous work relied on various versions of ImageNet to make such evaluations. But such datasets consist of images that are pretty likely to be encountered in a gigantic dataset such as LAION, making their evaluation less reliable. We believe that aiming for evaluation of the compositional generalization makes it possible to come up with test data with rare compositions, but with known constituents, that are extremely unlikely to be seen in the training sets. We collected a systematic and comprehensive list of such compositions, and did a thorough examination of the generated images through human inspection to make a dataset that helps the field to investigate OoD generalization more authentically. This also helped us shed light on one of the origins on OoD generalization in VLMs, which is the disentanglement at various levels.

---

### Author Response · Authors · 2023-11-20
**Looking forward to discussing with reviewers**

Dear reviewers,

Many thanks for your valuable feedback on our paper. We have gone through your points one-by-one and tried to address them carefully. We would appreciate if you could take a look at our response, and let us know if you have any further comments on the paper/rebuttal. We would be more than happy to take all your criticism, and incorporate them all in the paper.

Thanks!

---

### Comment · Area_Chair_FE77 · 2023-11-20
**Discussion between authors and reviewers**

Dear Reviewers,

Thanks for the reviews. The authors have uploaded their responses to your comments, please check if the rebuttal address your concerns and if you have further questions/comments to discuss with the authors. If the authors have addressed your concerns, please adjust your rating accordingly or vice versa.

AC

---

### Meta-Review · Area_Chair_FE77 · 2023-12-04

**Metareview:**

This paper investigates the cause of compositional generalization(CG) ability in the Vision-language models (VLMs). A new dataset, ImageNet-AO is proposed to benchmark the compositional capabilities of several CLIP models. Two main reasons are identified for improving CG: disentangled representation and large compositional training data.  Different representation disentanglement metrics are employed to validate the hypothesis.

Strengths:
+ The proposed dataset helps to evaluate compositional generalization in generative models. This could be significant to the community exploring model generalization.
+ Various analysis have been conducted to investigate the relationship between CG and representation disentanglement.

Weaknesses:
- It does not guarantee that the attributes in test set are not present or co-occur less in the training data.
- It lacks clarity in explaining motivation and justification of the proposed dataset.
- The conclusion is less convincing due to the limited candidates in each experiment.
- Some details are missing such as the context of 'switching dimension'  in 4.1, how human evaluation is conducted, etc.

**Justification For Why Not Higher Score:**

The idea proposed in this paper is interesting and insightful, and constructing a dataset based on compositionality is novel and has potential impact to the community. However, there are shared concerns by all reviewers on limited evaluation and missing details. I think the paper is not ready for publication at this stage, and the authors need more time to improve the paper.

**Justification For Why Not Lower Score:**

N/A

---

### Decision · Program_Chairs · 2024-01-16

Reject